# Myelinating Schwann cells and Netrin-1 control intra-nervous vascularization of the developing mouse sciatic nerve

Sonia Taïb[1], Noël Lamandé[1], Sabrina Martin[1], Fanny Coulpier[2], Piotr Topilko[2], Isabelle Brunet[1]*

[1]Center for Interdisciplinary Research in Biology (CIRB), College de France, CNRS, INSERM, Université PSL, Paris, France; [2]Institut Mondor de Recherche Biomédicale. UMR U 955 INSERM UPEC, Créteil, France

**Abstract** Peripheral nerves are vascularized by a dense network of blood vessels to guarantee their complex function. Despite the crucial role of vascularization to ensure nerve homeostasis and regeneration, the mechanisms governing nerve invasion by blood vessels remain poorly understood. We found, in mice, that the sciatic nerve invasion by blood vessels begins around embryonic day 16 and continues until birth. Interestingly, intra-nervous blood vessel density significantly decreases during post-natal period, starting from P10. We show that, while the axon guidance molecule Netrin-1 promotes nerve invasion by blood vessels via the endothelial receptor UNC5B during embryogenesis, myelinated Schwann cells negatively control intra-nervous vascularization during post-natal period.

## Editor's evaluation

This manuscript focuses on the cellular and molecular mechanisms underlying intra-nervous vascularisation of peripheral nerves during embryogenesis and early postnatal development. While the general molecular principles of angiogenesis and peripheral nerve development have been described, how these two processes are coordinated to form the intranervous vascular system is virtually unknown. Using mouse genetic models, the authors show that Schwann cells regulate vascularization of the sciatic nerve and are required for a decrease in vascular density postnatally.

*For correspondence:
isabelle.brunet@college-de-france.fr

**Competing interest:** The authors declare that no competing interests exist.

## Introduction

Formation of vascular plexus and its maturation into a dense network of blood vessels is one of the most important mechanisms during embryonic and post-natal development in vertebrates (*Kolte et al., 2015*). Among numerous physiological tasks, the vascular network transports oxygen and nutrients and eliminates waste products, all critical for organ survival and cellular homeostasis. During early stages of embryogenesis, by a process called vasculogenesis, a part of the mesoderm differentiates into endothelial cells, forms a lumen, and deposits a basal lamina to create a vascular plexus de novo. To become functional, blood vessels mature and specialize by recruitment of mural cells and form a network of arteries, capillaries, and veins. During development and in the adult, new blood vessels are formed from pre-existing ones by a process called angiogenesis. This mechanism involves specialized endothelial cells named tip cells. Thanks to the expression of specific receptors, tip cells can sense the microenvironment, and respond to guidance cues to lead the angiogenic sprout. Thus, angiogenesis not only allows the rapid vascularization of developing tissues and organs but ensures appropriate vascularization rate adapted to specific needs in nutrients and oxygen (*Carmeliet and Jain, 2011*).

Peripheral nerves, connecting the central nervous system (CNS) to the rest of the body, are composed of axons covered by myelinating and non-myelinating Schwann cells (SC). During neural development, axons are guided through embryonic tissues to reach their final targets (*Stoeckli, 2018*), supported by secreted trophic factors ensuring axonal survival until the connection is ultimately stabilized when the appropriate target is reached (*Ye et al., 2019*). Simultaneously, SC precursors (SCP) derive from the neural crest cells and migrate from the neural tube around embryonic day E10.5 to contact axons and differentiate into immature SC (iSC) around E15/E16 (*Woodhoo and Sommer, 2008*). Finally, around birth they differentiate into either myelinating or non-myelinating SC. SC are necessary to ensure axonal survival during development and also in regeneration (*Jessen et al., 2015*). Myelin sheaths are produced by SC wrapping larger axons in a 1:1 ratio to allow rapid saltatory conduction of action potentials. In rodents, myelination is progressive, starts around birth, and lasts during about 2 weeks (*Woodhoo and Sommer, 2008*). Non-myelinating SC associate with multiple small caliber axons to form Remak bundles without forming compact myelin sheaths. Adult nerves are stable structures, with nerve fibers surrounded and protected by three layers of connective tissues: the endoneurium, perineurium, and epineurium (*Kaplan et al., 2009*). While nerves acquire a complex, multi-cellular composition, especially with connective tissue, specific metabolic needs are covered by an adapted and specific vascularization. Indeed, a dense network called the *vasa nervorum* ensures the maintenance of proper nerve homeostasis. This is of particular importance when nerves are formed from axons covering a long distance from their soma to the innervated target. The organization of this vascular system has been previously described (*Boissaud-Cooke et al., 2015*). In fact, blood vessels are found within the three connective tissues, all along the nerve length, forming the peri- and intra (INV)-nervous vascular system. They are composed of a single layer of endothelial cells surrounded by mural cells: the SMC and pericytes allowing the contraction of the blood vessel to adapt the blood flow.

Whereas the different developmental stages of the axons, connective tissues and SC have been documented, it remains unclear when and how the *vasa nervorum* develops. A better understanding of the normal development of the peripheral nerves and their components, especially the *vasa nervorum*, is of crucial importance as nerves control many organs and tissues. Moreover, understanding the molecular control of nerve vascularization and re-vascularization after injury is of primary importance in regenerative medicine. In fact, after transection, peripheral nerves can repair and reconnect and this process implicates migration of SC to guide regrowing axons. It has been shown that blood vessels inside the nerve are used by SC to direct this migration (*Cattin et al., 2015*). Moreover, nerve grafts appears to be more efficient when their vascularization is preserved, allowing a better regeneration (*D'Arpa et al., 2015*).

In this study, we performed a time course analysis of the intra-nervous vascularization and discovered mechanisms for the development and maturation of the INV in the sciatic nerve. Importantly, we decipher the cellular and molecular actors governing the *vasa nervorum* formation.

## Results

### The INV develops rapidly from embryonic day E16

To define when the first blood vessels penetrate the nerves and mature into *vasa nervorum*, sciatic nerves were dissected and analyzed for the presence of blood vessels at successive stages starting from embryonic day 15 (E15). At this stage arteries and nerves start to be aligned in the skin (*Mukouyama et al., 2002*). Endothelial cells were visualized using CD31 marker in order to assess the level of nerve vascularization. At E15.5, the extrinsic artery of the primitive sciatic nerve is aligned with axons (TUJ-1 staining). However, no blood vessels were observed inside or surrounding the nerve (*Figure 1A*). At E16, we noticed new blood vessels emerging from the aligned extrinsic artery to form a peri-nervous vascular plexus (*Figure 1B*, *arrowhead*). Developing from this latter, numerous angiogenic sprouts invade the inside part of the immature nerve, starting to form the intra-nervous vascular network (INV) as observed in the optical section of the proximal part of the nerve (*Figure 1C*, *arrowhead*). However, at E16, in the distal part no blood vessels were found inside the nerve (*Figure 1D*), suggesting a proximo-distal gradient of nerve invasion by blood vessels. Soon after, at E16.5 (*Figure 1E*) and E17.5 (*Figure 1F*), we found more blood vessels inside the nerve. Later, at E19, many angiogenic sprouts are still found inside a dense vascular plexus, implying that angiogenesis is still ongoing (*Figure 1G*

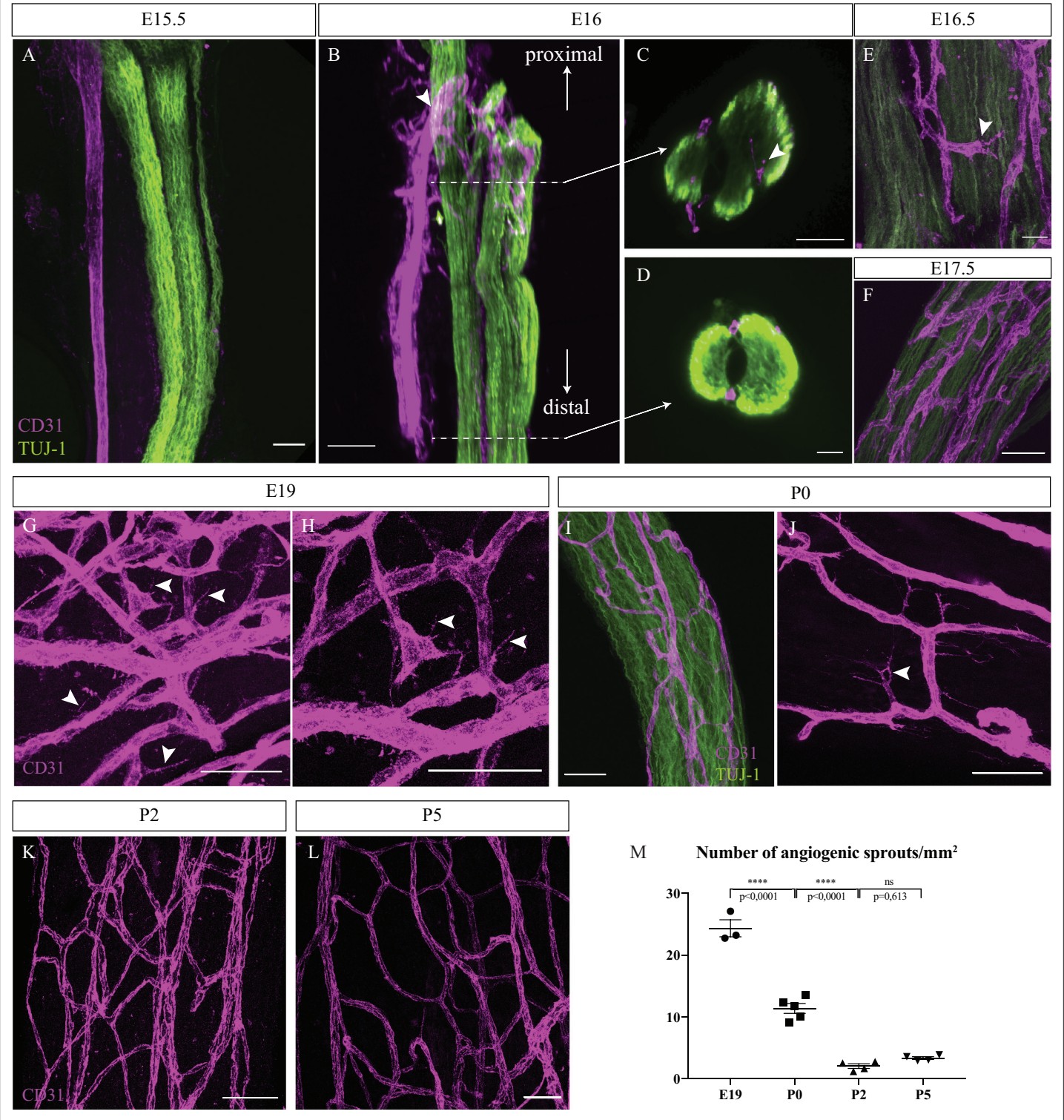

**Figure 1.** Intra-nervous vascular system (INV) develops rapidly starting from embryonic day E16 and angiogenesis is still ongoing at P0. (**A**) Whole-mount immunofluorescence staining of sciatic nerve showing axons (TUJ-1, green) and blood vessels (endothelial cells expressing CD31, magenta) at embryonic day E15.5. (**B**) Snapshot of a 3D view of primitive sciatic nerve at E16. New blood vessels (CD31, magenta) emerging from the aligned artery (arrowhead) starting to form the peri-nervous vascular system. (**C**) Orthogonal view of the proximal part of the nerve showing blood vessels starting to form the intra-nervous vasculature (arrowhead) and of the distal part (**D**) showing no intra-nervous blood vessels. (**E**) Close-up view of blood vessels inside a sciatic nerve at E16.5, showing several angiogenic sprouts with their filopodias (arrowhead). (**F**) INV of a sciatic nerve at E17.5. (**G**) INV of a sciatic nerve at E19, showing active angiogenesis with several angiogenic sprouts with their filopodias (arrowheads). (**H**) High-resolution image of angiogenic

*Figure 1 continued on next page*

*Figure 1 continued*

tip cells. (**I**) INV of a sciatic nerve at P0, with angiogenesis still ongoing (**J**), as angiogenic sprouts displaying filopodias are visible (arrowhead). (**K, L**) At P2 and P5, almost no angiogenic sprouts are visible. Around six nerves from six mice were analyzed per developmental stage. (**M**) Quantification of the number of angiogenic sprouts per 1 mm$^2$ throughout peri-natal development. n = 3–5 animals for each stage, one or two nerve(s) per animal, mean ± SEM, one-way ANOVA and Tukey's multiple comparisons test, ****p < 0.0001. Scale bars are 50 µm for all images except for (**J**), the scale bar is 20 µm. Detailed values are presented in *Figure 1—source data 1*.

The online version of this article includes the following source data for figure 1:

**Source data 1.** Quantification of the number of angiogenic sprouts.

*and H, arrowheads*). At birth (P0) (*Figure 1I*), angiogenic sprouts were also found inside the nerve (*Figure 1J, arrowheads*). Finally, at P2 and P5, very few angiogenic sprouts were observed (*Figure 1K and L*). This level of angiogenesis was quantified by counting the number of angiogenic sprouts found inside the nerve at different stages during peri-natal development. We found that the number of angiogenic sprouts per mm$^2$ significantly decreases throughout development, between E19 and P0 and between P0 and P2, to reach a number of almost zero angiogenic sprout at P5 (*Figure 1M*).

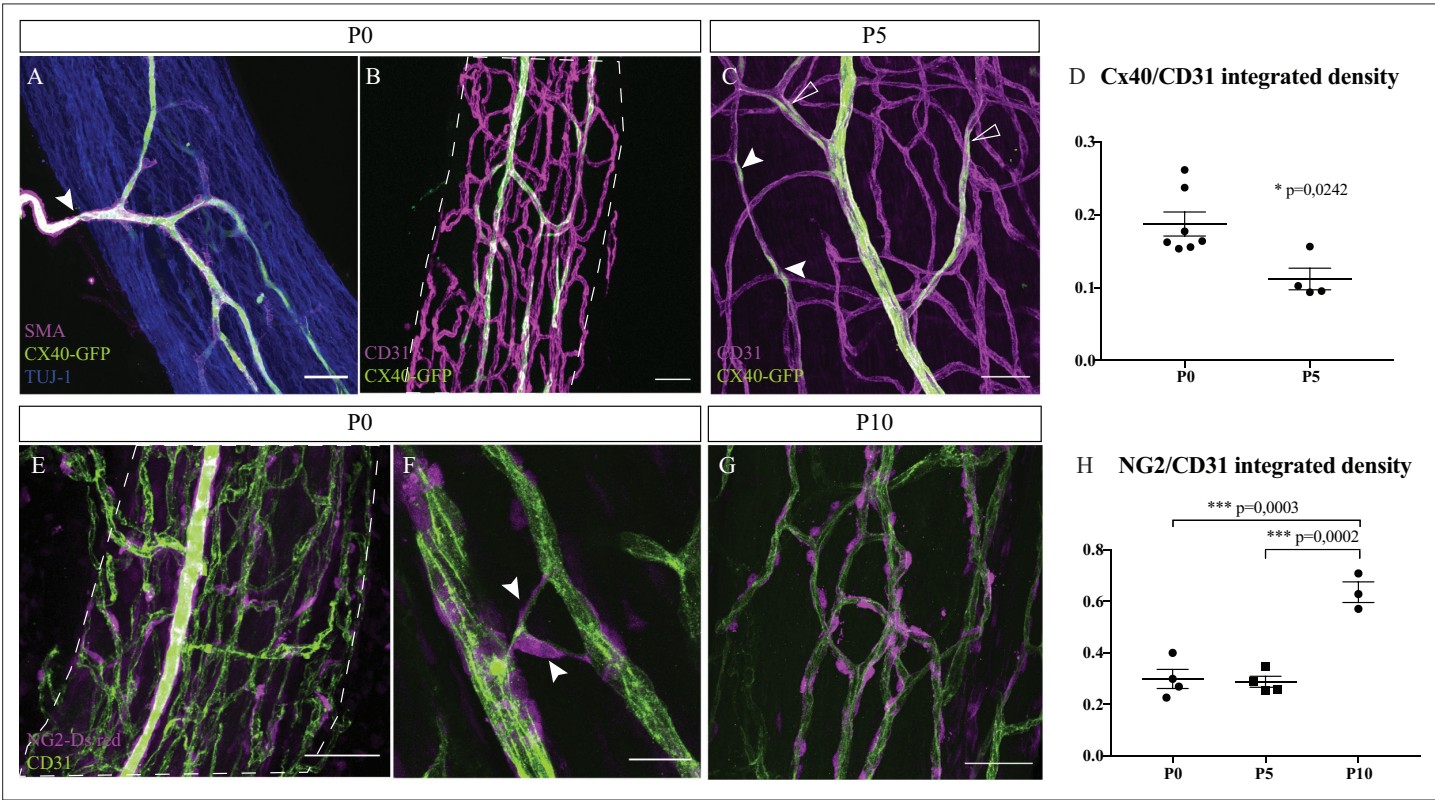

**Figure 2.** Intra-nervous vascular system (INV) matures during post-natal stages with arterial differentiation and pericytes recruitment. (**A**) Whole-mount immunofluorescence staining of a sciatic nerve from a connexin-40 (Cx-40)-GFP reporter mouse at P0 showing an artery composed of endothelial cells expressing the protein Cx-40 (in green) and covered by smooth muscle cells (stained in magenta, smooth muscle actin staining [SMA]). This artery has invaded the sciatic nerve (axons expressing TUJ-1, blue). (**B**) INV at P0 is marked in magenta and endothelial cells expressing the protein Cx-40 are in green. (**C**) At P5, some endothelial cells express the protein Cx-40 (white arrowheads) and new arterial branches are differentiating (empty arrowheads). (**D**) Quantification of the ratio of Cx-40-GFP-positive endothelial cells at P0 and P5. N = 4–7 nerves from four to seven different animals; mean ± SEM, Mann-Whitney test. (**E**) NG2-Dsred mice were used to observe pericytes (magenta) covering endothelial cells (CD31, green) of a P0 sciatic nerve. (**F**) Close-up view of P0 sciatic nerve vasculature showing pericytes covering new blood vessel branching (arrowheads). (**G**) Pericytes coverage of the INV at P10. Scale bars are 50 µm for A, B, C, E, and 20 µm for D. Around six nerves were analyzed per developmental stages. (**H**) Quantification of NG2-DsRed-positive cells covering blood vessels at P0, P5, and P10. N = 3–4 nerves from three to four different animals; mean ± SEM, one-way ANOVA and Tukey's multiple comparisons test, ***p < 0.001. Detailed values for C and G are presented in *Figure 2—source data 1*.

The online version of this article includes the following source data for figure 2:

**Source data 1.** Quantification of Cx-40-GFP+ and NG2-Dsred+ coverage.

## Blood vessels inside the nerve mature during post-natal development, undergoing arterial differentiation and pericytes recruitment

To be fully functional, blood vessels have to be mature and organized as a hierarchical vascular network. Endothelial cells of arterioles specifically express the protein connexin-40 (Cx-40), as observed at P0 (*Figure 2B*). These branches are also covered by smooth muscles cells (SMC) to control their diameter and adapt the blood flow. The extrinsic artery, expressing Cx-40, visualized by GFP expression, and covered by SMC (*Figure 2A, arrowhead*) is divided into continuous ascending and descending arterioles to directly supply each region of the nerve and ensure arterial blood flow to the entire nerve. Pre-existing blood vessels of the INV also undergo arterial differentiation, as observed in sciatic nerve at P5. Endothelial cells expressing Cx-40 in an isolated manner were visible (*Figure 2C, arrowheads*), suggesting that new arterial branches undergo differentiation inside the nerve. Notably, arterialization of the penetrating vessels is still ongoing as some parts are already covered by SMC whereas newly differentiated arteries have still not recruited SMC (*Figure 2A*). The Cx-40/CD31 integrated density was higher in nerves at P0 compared to P5 (*Figure 2D*). Moreover, pericytes, expressing the protein NG2 and covering endothelial cells, are important for maintenance of the blood-nerve barrier and are able to contract to change the capillary diameter (*Bergers and Song, 2005*). At P0, numerous endothelial cells of the INV are already covered by pericytes (*Figure 2E*). Some of these cells cover new blood vessel branching (*Figure 2F*) as it has been already reported that pericytes are critical for angiogenesis (*Bergers and Song, 2005*). These pericytes at P0 have an elongated shape and their coverage is relatively sparse compared to the pericytes observed at P10 (*Figure 2G*). Blood vessel coverage by pericytes NG2+ cells increase between P5 and P10 (*Figure 2H*) suggesting maturation and stabilization of the new blood vessels whereas angiogenesis is almost stopped. Thus, at birth, INV already displays vessel wall maturation that will remain during nerve post-natal morphogenesis.

## INV density decreases during post-natal development

Post-natal maturation of the peripheral nerves includes the production of the myelin sheaths by SC. Myelination starts at birth, continues for about 2–3 weeks, and increases both nerve size and caliber (*Jessen and Mirsky, 2005*). We hypothesized that the growth of the nerve would provoke an increase of the INV during post-natal development to adapt the nerve supply in oxygen and nutrients. To test this hypothesis, we analyzed CD31 immunoreactivity in sciatic nerves cross sections from P0 (*Figure 3A*) to young adult stage (8-week-old mice) (*Figure 3B*).

As expected, the caliber of the sciatic nerve gradually increases during post-natal development by a factor of 10 from P0 to young adulthood (*Figure 3C*). Area density of CD31 immunoreactivity was quantified as percentage of total nerve area and appears to vary over time (*Figure 3D*). The intranervous CD31-positive area covers around 8% of the nerve at P0. This value decreases after birth significantly at P10, to reach a stable coverage of approximately 2% of the nerve, and does not change later at young adult stage. This result corroborates the data we obtained about the progressively decreasing number of angiogenic tip cells during post-natal development (*Figure 1I*). Interestingly, this decrease in INV rate correlates with the myelination period during the 2 weeks after birth (gray box, *Figure 3D*).

## Ablation of myelinating SC leads to an abnormal vascularization of the sciatic nerve

As nerve vascularization density decreases during post-natal development and while myelin is produced and since the majority of cells within the sciatic nerve are SC (~70%) (*Stierli et al., 2018*), we wondered if myelinating SC could affect nerve vascularization. It has been previously described that SC promote endothelial cells migration in vitro (*Ramos et al., 2015*). To address in vivo the question of whether these cells have a role in the development of the INV, we used a transgenic mouse line in which genetic ablation of SCP was provoked. The transcription factor KROX20, expressed by immature SC, represents the master regulator of myelin genes and thus controls myelination formation and maintenance (*Topilko et al., 1994*). We crossed *Wnt1-Cre/+* mice (*Joseph et al., 2004*) with *Krox-20GFP(DT)/+* mice (*Vermeren et al., 2003*) to obtain *Wnt1-Cre;Krox20GFP(DT)/+* pups in which the A chain of the diphtheria toxin (DT) is only expressed upon cre-mediated recombination, to lead to a specific death of immature SC expressing Krox20 around E15/E16. In fact, at E19, there is significantly less SC (expressing SOX10) in sciatic nerves from *Wnt1-Cre;Krox20GFP(DT)/+* mutant mice (*Figure 4—figure*

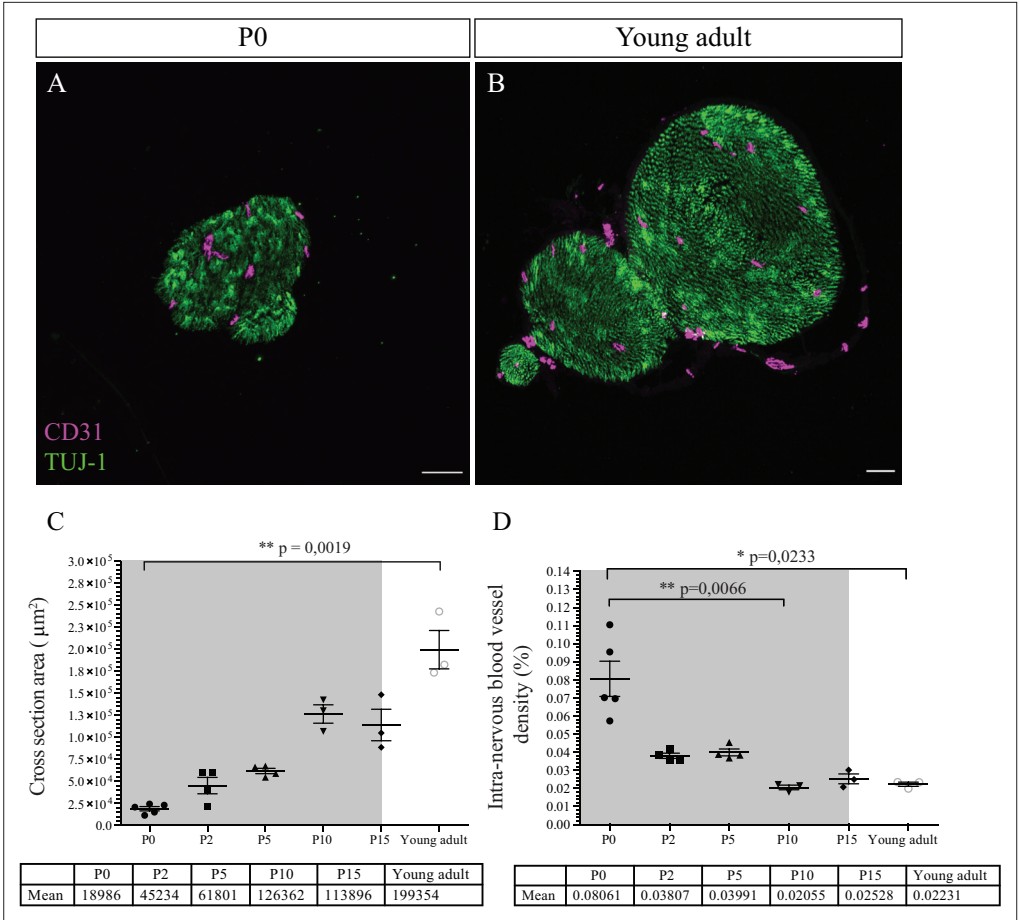

**Figure 3.** Intra-nervous vascular system (INV) density decreases during post-natal development. (**A**) Immunofluorescence staining of sciatic nerve's transversal sections from a P0 mouse and from a young adult mouse (**B**) showing blood vessels (CD31, magenta) and axons (TUJ-1, green) inside the nerve. (**C**) Quantification of the surface area of nerves' cross section, from P0 to young adult stage (around P54). The gray box represents the myelination period from P0 to P15. (**D**) Intra-nervous CD31 density expressed as percentage of total nerve cross section area. Only the blood vessels inside the nerve were counted. Blood vessel's density decreases significantly from P0 to P10 and stabilizes. n = 3–5 animals for each stage, one nerve per animal and >50 cross sections were analyzed per nerve, mean ± SEM, the comparison was made between all stages (two by two using multiple comparison) and only the significant differences were showed: Kruskal-Wallis and Dunn's multiple comparisons test, *p < 0.05, **p < 0.01, only. Detailed values are presented in **Figure 3—source data 1**.

The online version of this article includes the following source data for figure 3:

**Source data 1.** Quantification of intra-nervous vascular system (INV) density.

---

supplement 1A, A', B and B') and at P5, less myelinating SC (normally expressing S100β protein) (**Figure 4—figure supplement 1C and C'**). These animals survive only few days after birth. Therefore, we analyzed the level of vascularization of sciatic nerves at P5. As expected, the sciatic nerves of *Wnt1-Cre;Krox20*^{GFP(DT)/+} mice were thinner as compared to *Wnt1-Cre* control littermates (**Figure 4A and B**). This was due to the lack of SC and their myelin sheaths, which normally considerably increase the nerve's caliber. Furthermore, the INV network is disrupted, denser, and more anastomotic, resembling embryonic immature plexus (**Figure 4C**). To further analyze the consequences of the SC ablation on the INV, we performed 3D reconstruction of this vasculature in toto (**Figure 4D and E**). We quantified the total length of the vasculature tree in blue and the number of blood vessel branching points in red. These values were normalized to the nerve size which was found to be significantly different between the two groups (**Figure 4F**). Sciatic nerves from mutant pups display an increased total length of blood vessel network (**Figure 4G**) with more branches (**Figure 4H**) as compared to the control littermates. To make sure that the phenotype we observed was not a consequence of decreased nerve size, we

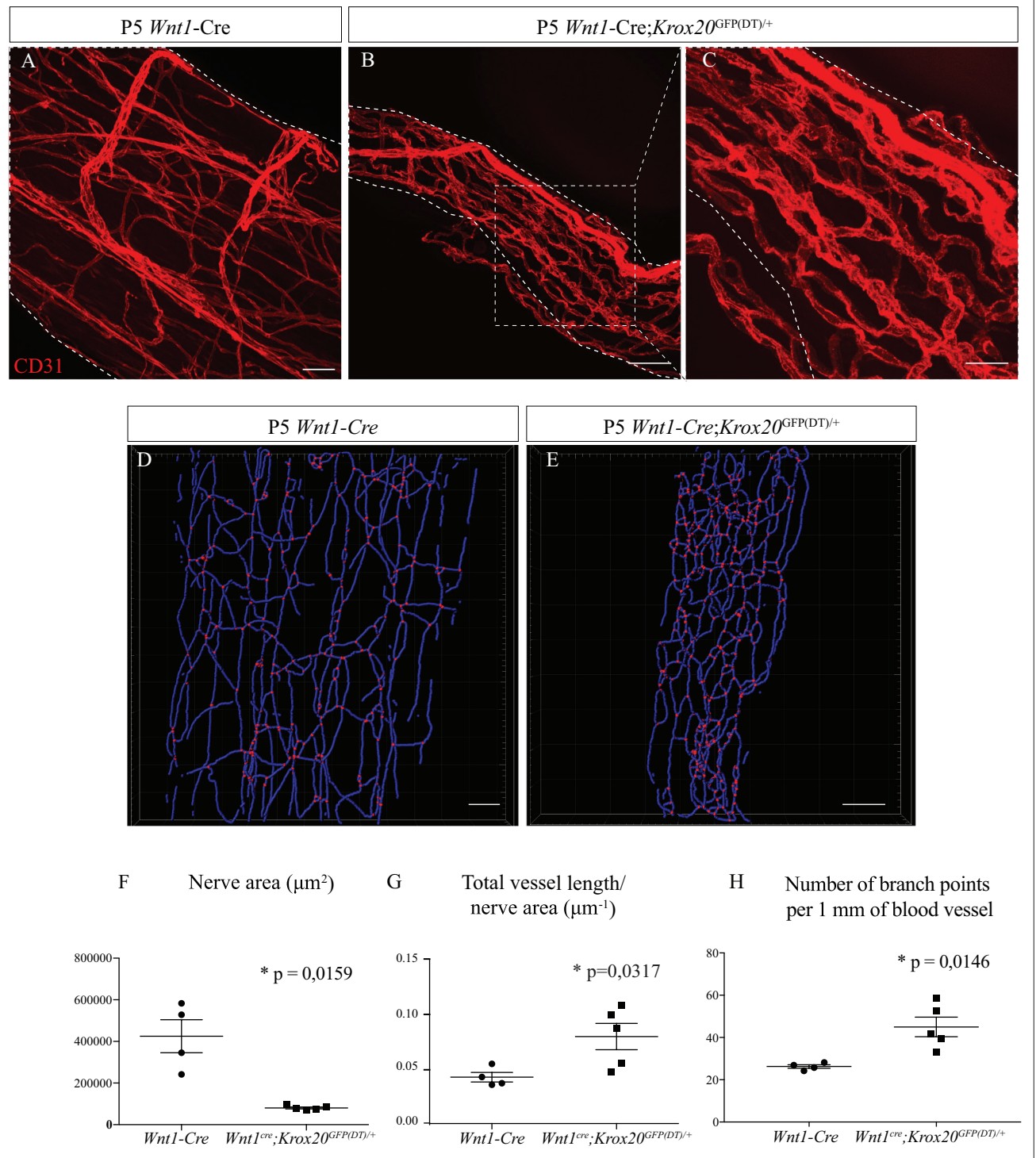

**Figure 4.** Schwann cells ablation disrupt sciatic nerve vasculature. (**A**) Whole-mount immunofluorescence staining of sciatic nerve from control *Wnt1*-Cre mouse at P5, the blood vessels are stained with CD31 and shown in red. (**B**) Vasculature of a sciatic nerve from a *Wnt1*-Cre;*Krox20*<sup>GFP(DT)/+</sup> mouse at P5. (**C**) Close-up view showing the disruption of the blood vessels' organization. (**D, E**) *Imaris* 3D reconstruction of sciatic nerve vasculature in blue, the branching points are shown in red. (**F**) Quantification of sciatic nerve area. (**G**) Quantification of total blood vessel length/nerve area. (**H**) Number of branching points per 1 mm of blood vessel (n = 4–5 per group, graphs show mean ± SEM, Mann-Whitney test, *p < 0.05). Scale bars are 50 μm for A, B, D, and E and 20 μm for C. Detailed values are presented in *Figure 4—source data 1*.

The online version of this article includes the following source data and figure supplement(s) for figure 4:

**Source data 1.** Quantification of nerve area and intra-nervous vascular system (INV) length and branch points in *Wnt1*-Cre and *Wnt1*-Cre;*Krox20*<sup>GFP(DT)/+</sup>

*Figure 4 continued on next page*

*Figure 4 continued*

mice.

**Figure supplement 1.** Schwann cells (SC) genetic ablation in Wnt1-Cre;Krox20GFP(DT)/+ mutant mice.

**Figure supplement 1—source data 1.** Quantification of nerve area and sox10+ area in *Wnt1*-Cre and *Wnt1*-Cre;*Krox20*GFP(DT)/+ mice.

**Figure supplement 2.** Intra-nervous vascular system (INV) disruption after Schwann cells genetic ablation in Wnt1-Cre;Krox20GFP(DT)/+ mutant mice.

**Figure supplement 2—source data 1.** Quantification of nerve area and CD31+ area in *Wnt1*-Cre and *Wnt1*-Cre;*Krox20*GFP(DT)/+ mice.

also analyzed the INV of sciatic nerves from *Wnt1-Cre;Krox20*$^{GFP(DT)/+}$ mice but at P0 (at the beginning of myelination) (*Figure 4—figure supplement 2A and A'*). At this stage, nerve size is not different between the two groups, whereas vasculature is disrupted and more dense in the sciatic nerves of *Wnt1-Cre;Krox20*$^{GFP(DT)/+}$ mice (*Figure 4—figure supplement 2B and B'*). Altogether, these results suggest that myelinating SC could control INV development and maturation probably by negatively controlling angiogenesis in order to ensure a proper amount of blood vessels inside the nerve.

## Hypervascularization of the sciatic nerve following inhibition of myelin production

To decipher the role of the myelin sheath, independently of SC, we used the *Krox20*$^{Cre/Fl}$ mouse line. As previously described, in these mice, *Krox20* from the floxed allele is expressed until enough Cre recombinase accumulates resulting in the delayed inactivation of the *Krox20* gene in myelinating SC (*Decker et al., 2006*). Indeed, *Krox20* gene is normally activated at E15 but the protein is detectable shortly after birth (*Figure 5—figure supplement 1A*). Thus, general organization of the mutant sciatic nerve, including proportion of iSC, is not affected between E15.5 and E18.5. At P1, Krox20 protein is fairly expressed but there is no difference between *Krox20*$^{Cre/Fl}$ mice and wild-type (WT) littermates (*Decker et al., 2006*). The level of Krox20 protein progressively decreases in the mutant (50% at P4) to become undetectable at P28 and beyond. Thus, during embryonic period of the nerve development, in particular at the stages when INV is initiated (E16.5), sciatic nerve remains unaffected. Therefore, development of the sciatic nerve of *Krox20*$^{Cre/Fl}$ mice is unaffected until birth and thus early angiogenesis events leading to INV formation should not be affected in the mutant. *Krox20* inactivation blocks SC at an early stage of their differentiation, progressively prevents the formation of myelin sheaths (*Figure 5A*) and this phenotype is maintained through the 3 weeks of survival observed for those mice (*Decker et al., 2006*). Indeed, *Krox20*$^{Cre/Fl}$ mutants survive around 20 days after birth and show tremors. Besides a complete absence of the myelin sheath, *Krox20* deletion in promyelinating Schwan cells does not alter any other property in the post-natal sciatic nerve, with a normal axonal density, as already described (*Decker et al., 2006*). We chose to analyze sciatic nerves before lethality, at P19, so that the sciatic nerves from control pups are fully myelinated whereas the ones from *Krox20*$^{Cre/Fl}$ mice are devoid of myelin. The vasculature of sciatic nerves from control and mutant mice were compared using CD31 staining on whole-mount preparations. Interestingly, sciatic nerves of mutant mice appear to be more densely vascularized compared to control littermates (*Figure 5B and B'*). Thus, we performed 3D reconstruction of the INV (*Figure 5C and C'*). As compared to control mice, quantifications revealed a similar nerve size (*Figure 5D*) but higher blood vessel total length (*Figure 5E*) and higher number of branching points in mutant's sciatic nerves (*Figure 5F*). We then aimed to better characterize vascular network defects and assessed maturation and permeability of the INV. Endothelial cells composing the INV express claudin-5, a tight junction protein, important for the acquisition of a barrier property (*Peltonen et al., 2013*). At P18, in sciatic nerves from control mice, claudin-5 staining colocalized with CD31 staining in almost all blood vessels (*Figure 5G*). In sciatic nerves from *Krox20*$^{Cre/Fl}$ mice, claudin-5 is poorly expressed by endothelial cells as almost all blood vessels branches are claudin-5 negative (*Figure 5G'*, *arrowheads*). Altogether, those data suggest that myelin sheath controls sciatic nerve vascularization by inhibiting angiogenesis and by limiting vascular rate and maturation.

Interestingly, the number of proliferating cells expressing Ki67 is considerably higher in the unmyelinated sciatic nerves (*Figure 5H,H',I*). As reported before (*Topilko et al., 1994*), these proliferating cells are SC, as they specifically express the protein SOX10 (*Figure 5J*). Indeed, activation of the genes controlling myelination, such as KROX20, induces the arrest of the proliferation state of SC in favor of differentiation and production of myelin sheath (*Salzer, 2015*).

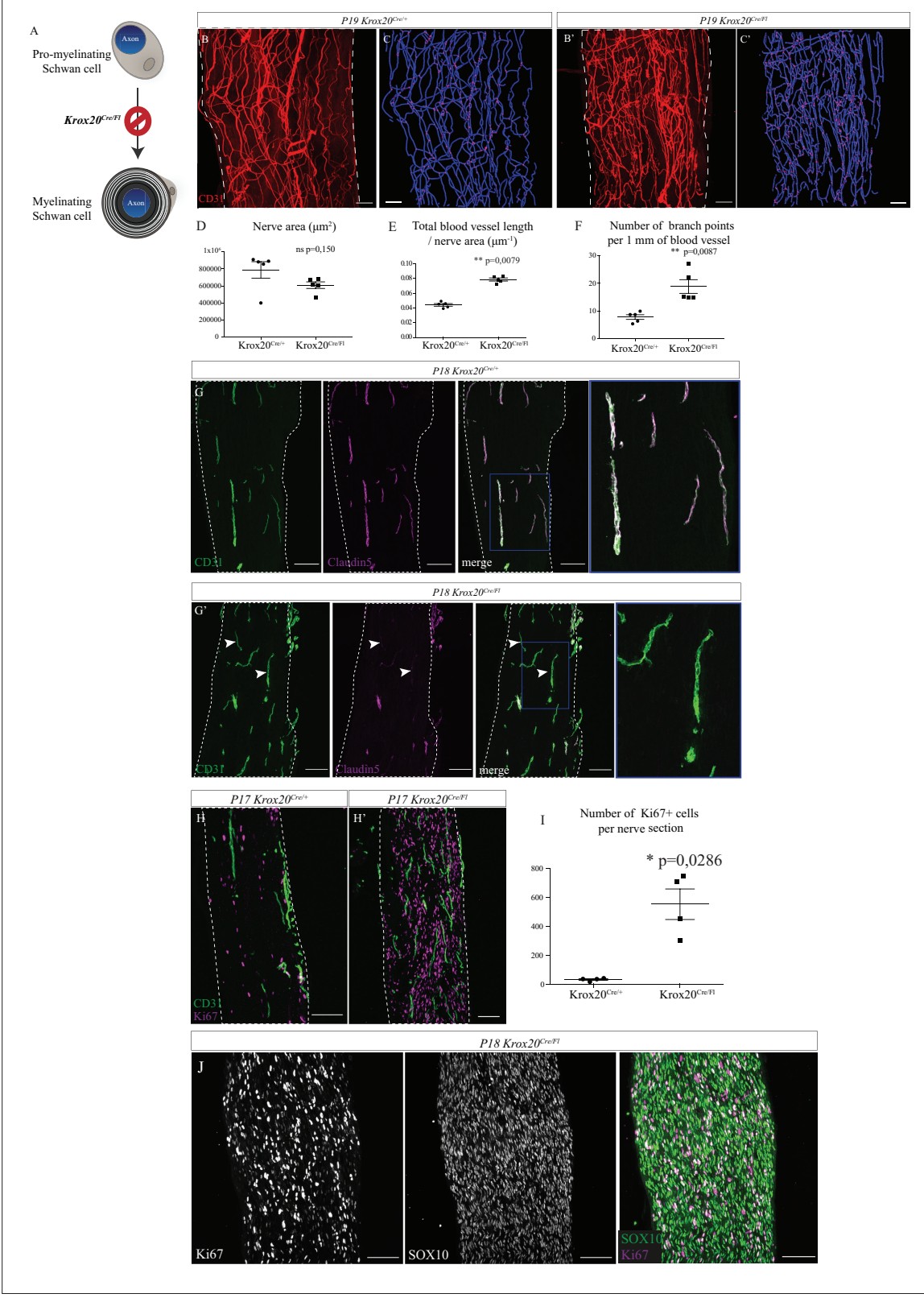

**Figure 5.** Sciatic nerve is hypervascularized upon myelination inhibition. (**A**) Schematic illustrating *Krox20* knock-out leading to myelination inhibition. (**B, B'**) Whole-mount immunofluorescence staining of sciatic nerves from a *Krox20*[Cre/+] control mouse and from a *Krox20*[Cre/Fl] mouse at P19, blood vessels are stained in red. (**C, C'**) Imaris 3D reconstruction of sciatic nerve vasculature in blue, the branching points are in magenta. (**D**) Quantification of the sciatic nerves area in µm². (**E**) Quantification of the length of the entire vasculature in µm. (**F**) Quantification of the number of branching points

*Figure 5 continued on next page*

*Figure 5 continued*

per 1 mm of blood vessel (n = 5 different nerves from five different animals, per group; graphs show mean ± SEM, Mann-Whitney test, **p < 0.01). (**G, G'**) Longitudinal sections of sciatic nerve dissected *Krox20^{Cre/+}* and *Krox20^{Cre/Fl}*, at P18. Endothelial cells expressing CD31 are in green and claudin-5 in magenta. Dotted lines represent the borders of the nerve section. Some vessel branches do not express claudin-5 (arrowheads) (**H, H'**) Longitudinal sections of sciatic nerves dissected from a *Krox20^{cre/+}* mouse and *Krox20^{cre/Fl}*, at P17. Cells in proliferation express the protein Ki67 (magenta) and blood vessels are in green. (**I**) Quantification of Ki67-positive cells per nerve section. n = 4 animals per group, more than 25 sections were quantified per nerve, graph shows means ± SEM, Mann-Whitney test, *p < 0.05. (**J**) SOX10 and Ki67 staining on sciatic nerve longitudinal sections. Scale bars are 100 μm. Detailed values are presented in *Figure 5—source data 1*.

The online version of this article includes the following source data and figure supplement(s) for figure 5:

**Source data 1.** Quantification of nerve area, intra-nervous vascular system (INV) length, and branch points, Ki67+ area in *Krox20^{Cre/Fl}* mice.

**Figure supplement 1.** Krox20 protein is detectable after birth in the sciatic nerve.

## The guidance molecule Netrin-1 and the receptor UNC5B are involved in INV formation

We next investigated the molecular control of INV development at embryonic stages. Netrin-1 is a guidance molecule known to control axonal guidance and also angiogenesis (*Boyer and Gupton, 2018*; *Bradford et al., 2009*; *Park et al., 2004*). Studies have shown that it can also be implicated in nerve regeneration, stimulating axonal and blood vessels regrowth (*Dun and Parkinson, 2017*; *Madison et al., 2000*). Therefore, we asked whether Netrin-1 could regulate sciatic nerve vascularization during normal development. As we described, vascularization of sciatic nerves is still ongoing around birth, involving active angiogenesis. First, we used *Ntn1^{lacZ/+}* knock-in mice to report Netrin-1 protein expression in the nerve. At E16, β-galactosidase expression was found in different regions of the embryo's limb, but not inside the sciatic nerve (*Figure 6—figure supplement 1A*). At E17, a stage during which new blood vessels are attracted into the endoneurial part of the nerve, we noticed that β-galactosidase expression was found inside and surrounding the nerve (*Figure 6—figure supplement 1B* and B'). At P2, β-galactosidase activity, reporting Netrin-1 expression, was found close to blood vessels, in areas compatible with a potential role of Netrin-1 in endothelial cell guidance (*Figure 6A*). Single cell transcriptomic analysis of the different cell types constituting sciatic nerves showed that *Ntn1* is in fact expressed by epineurial and perineurial cells and proliferating fibroblast-like cells (*Gerber et al., 2021*). We confirmed at P0 that expression of β-galactosidase (reporting Netrin-1 expression) was found at the level of FAP+ cells (fibroblast activation protein) located at the epi-perineurial cell layer (*Figure 6B and Figure 6—figure supplement 1C*). Using FISH (fluorescent in situ hybridization), we also found *Ntn1* mRNA expressed by epi-perineural cells expressing *Fap* mRNA (*Figure 6C, C'* and *Figure 6—figure supplement 1D*).

As *Ntn1^{-/-}* mice die around birth (*Serafini et al., 1996*), we analyzed entire sciatic nerves of *Ntn1^{lacZ/lacZ}* knock-in embryos at E16, before INV onset (*Figure 6—figure supplement 1E and E'*). We found no difference in terms of sciatic nerve diameter (*Figure 6—figure supplement 1F*), suggesting that the lack of Netrin-1 did not have a significant impact on sciatic nerve development and thickness of the segment analyzed in our study (*Figure 6—figure supplement 2A*), reflecting no major axonal guidance defect nor defasciculation at this sciatic nerve level at that developmental stage. We then explored sciatic nerves vascularization of embryos at E17.5, when INV has already initiated. Embryonic sciatic nerves were dissected and we assessed blood vessels quantity using CD31 staining (*Figure 6—figure supplement 1G and G'*). Whereas there is no difference in terms of nerve area (*Figure 6D*), we found that nerves from mutant embryos are less vascularized as compared to control embryos (*Figure 6E*). To confirm this result, and to make sure that the dissection did not alter the nerve integrity and vascularization, we quantified nerve vasculature at this stage directly within the embryonic limb. Embryos' limbs were entirely dissected, as they contain the sciatic nerve (*Figure 6F and F'*, left top corner, dotted area). We performed whole-mount immunostaining of the entire limb to stain axons (neurofilament [NF]) and blood vessels (CD31). Limbs were then cleared using an adapted iDISCO+ protocol (*Renier et al., 2014*) and the areas of interest were imaged in three dimensions. Using *Imaris* software, we observed the sciatic nerve inside the limb, thus keeping the physical integrity of the nerve and its surrounding (*Figure 6G and G'*). We also found that, whereas sciatic nerves of *Ntn1^{lacZ/lacZ}* embryos had the same area as control littermates (*Figure 6H*), they were less vascularized. This was confirmed by quantification of CD31-positive area density inside the nerve (*Figure 6I*), with

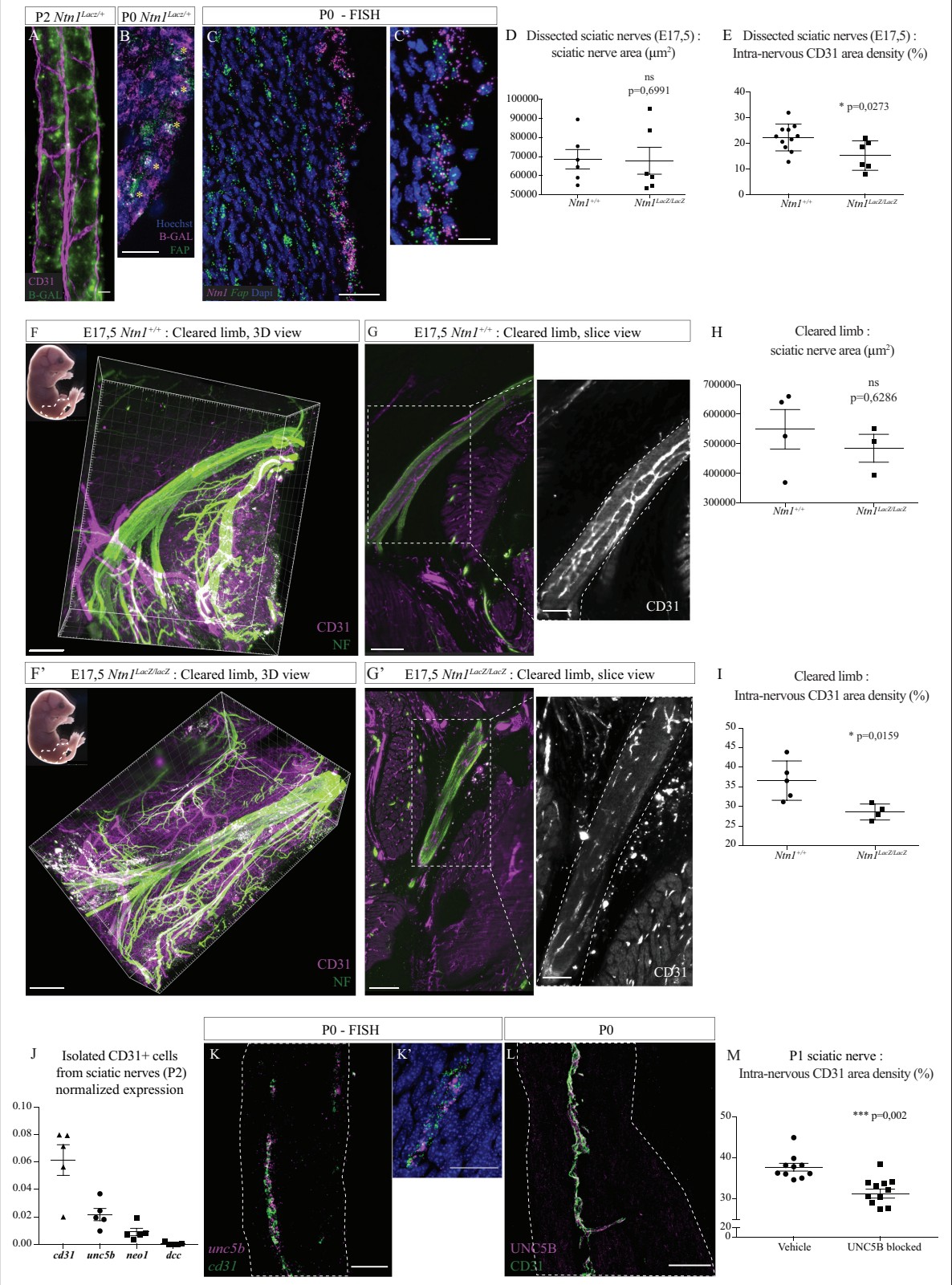

**Figure 6.** Netrin-1 and UNC5B control intra-nervous vascularization. (**A**) Whole-mount X-Gal staining (showed in green) of sciatic nerve from *Ntn1^Lacz/+^* mouse at P2 and vasculature is showed in magenta (CD31). (**B**) Immunofluorescence staining of β-galactosidase (magenta) and FAP (fibroblast-activating protein, green) on transversal cryo-section of sciatic nerve from *Ntn1^Lacz/+^* mouse at P0. Nuclei are stained with Hoechst. (**C, C'**) Fluorescent in situ hybridization (FISH) of *Ntn1* (magenta) and *Fap* (green) mRNA on longitudinal section of sciatic nerve from a wild-type (WT) mouse at P0 (**C**) and nuclei

*Figure 6 continued on next page*

*Figure 6 continued*

are labelled with Dapi. Close-up view of *Ntn1* high expression by epi-perineural cells expressing *Fap* (**C'**). (**D**) Quantification of area (from neurofilament [NF] staining) of sciatic nerves from *Ntn1⁺/⁺* and *Ntn1^LacZ/LacZ* embryos at E17.5. (**E**) Quantification of the intra-nervous CD31-positive total area after images thresholding, showing that vascularization is significantly reduced in mutant embryos. (**F, F'**) Top left corner: E17.5 embryos (the dotted line represents the dissected limb containing the primitive sciatic nerve) and 3D view from *Imaris* software of the embryo's limb, after CD31 (magenta) and NF (green) staining and clearing protocol (iDISCO+). Dotted lines delimit the immature sciatic nerve. (**G, G'**) Slice view of the cleared limb allowing the observation of the blood vessels inside the sciatic nerve. (**H**) Quantification of nerve area (from NF staining). (**I**) Quantification of intra-nervous CD31-positive area, using optical slices from cleared E17.5 embryos' limbs. n = 4–5, mean ± SEM, Mann-Whitney test, *p < 0.05. (**J**) Normalized expression of mRNA coding *cd31* (endothelial marker) and Netrin-1 receptors: *unc5b*, *neo1*, and *dcc* by isolated endothelial cells from sciatic nerves from P2 mice. N = 5 tubes containing around 1.17 × 10⁴ cells purified from 10 pooled sciatic nerves. (**K, K'**) FISH of *unc5b* (magenta) and *cd31* (green) mRNA on longitudinal section of sciatic nerve from WT mouse at P0 (**K**). Close-up view of blood vessel expressing *unc5b* and *cd31* mRNA (**K'**). (**L**) Immunofluorescence staining of UNC5B (magenta) and CD31 (green) on longitudinal section of sciatic nerve from WT mouse at P0. (**M**) Quantification of the intra-nervous CD31-positive total area from mice 1 day after they were injected with anti-UNC5B antibodies at P0 (dose = 4 μg/g of body weight). N = 10 and 11 pups were analyzed. Scale bar is 20 μm for A, B, and C', 50 μm for C, K, K', and L, 200 μm for F and F'. Detailed values are presented in *Figure 6—source data 1*.

The online version of this article includes the following source data and figure supplement(s) for figure 6:

**Source data 1.** Quantification of nerve area and CD31+ area in *Ntn1⁺/⁺* and *Ntn1^LacZ/LacZ* embryos; quantification of level of expression of *unc5b*, *neo1*, and *dcc* by isolated endothelial cells from sciatic nerves from P2 mice; quantification of CD31+ area in sciatic nerves from pups injected with anti-UNC5B antibodies at P0.

**Figure supplement 1.** Netrin-1 expression and sciatic nerve size from *Ntn1^LacZ/LacZ* mice.

**Figure supplement 1—source data 1.** Quantification of the average diameter of sciatic nerve from *Ntn1^LacZ/LacZ* and *Ntn1⁺/⁺* embryos.

**Figure supplement 2.** Sural, phrenic, intercostal, and cutaneous nerves are devoid of intra-nervous vascular system (INV).

a similar ratio and in agreement with our findings on dissected nerves. Interestingly, when we looked at the sural nerve, a purely sensory branch of the sciatic nerve (***Figure 6—figure supplement 2A***), we found no blood vessels at this stage in neither *Ntn1^lacZ/lacZ* nor *Ntn1⁺/⁺* mice (***Figure 6—figure supplement 2B and B'***). Other nerve types such as phrenic nerve, intercostal (mixed motor/sensory), or cutaneous (purely sensory nerve) were also devoid of INV at peri-natal stages (E17.5-P2) (***Figure 6—figure supplement 2C to E''***).

As Netrin-1 controls angiogenesis via its receptors expressed by endothelial cells (***Castets and Mehlen, 2010***), we checked whether endothelial cells from sciatic nerves express known Netrin-1 receptors: UNC5B, DCC, and neogenin1. To do so, we isolated endothelial cells from sciatic nerves of mice at P2 and performed quantitative PCR (qPCR). We found that endothelial cells from INV do not express *dcc* mRNA, whereas *neogenin1* is weakly expressed and *unc5b* is highly expressed (***Figure 6J***). Using FISH technique, we established that *unc5b* is indeed expressed by endothelial cells of INV of sciatic nerve at P0 (***Figure 6K and K'***, ***Figure 6—figure supplement 1H***). Finally, we also confirmed that UNC5B protein is expressed by those endothelial cells (***Figure 6L*** and ***Figure 6—figure supplement 1I***).

In order to address the role of the receptor UNC5B regarding the development of the INV, we injected pups (at P0) with blocking antibodies targeting specifically UNC5B and analyzed the level of vascularization of sciatic nerves 1 day after injection (***Figure 6—figure supplement 1J and J'***). We found that INV density is significantly lower in sciatic nerves from pups injected with anti-UNC5B compared to pups injected with the vehicle (***Figure 6M***).

These data suggest that Netrin-1, expressed at the level of epi-perineurial layer, and its receptor UNC5B, expressed by endothelial cells, are likely to be regulating the attraction and angiogenesis of the blood vessels inside the sciatic nerve around birth.

## Discussion

In the developing peripheral nerves, blood vessel formation is tightly controlled, both spatially and temporally, as the INV has to be well adapted to maintain homeostasis and ensure proper function. This study provides the time course of the development and maturation of the mouse sciatic nerve's INV. We find that INV appears relatively late during embryonic development, around E16, as compared to the vasculature of the CNS which begins around E9.5 (***Tata et al., 2015***; ***Himmels et al., 2017***) to ensure adequate delivery of oxygen and nutrients to neural progenitors. This temporal difference

lies in the specific timing of CNS morphogenesis that starts earlier and matures over a longer time. E16 in the mouse embryo represents a turning point for the peripheral nervous system (PNS) as SCP acquire an immature phenotype (*Jessen and Mirsky, 2019*). Interestingly, this step matches the apparition of the first blood vessels inside the nerve, suggesting there may be different molecular cues, expressed by immature SC participating in the attraction of blood vessels into the nerve. Another interesting observation is that other nerve types such as phrenic, intercostal (mixed motor/sensory), cutaneous, and sural nerve (purely sensory branch of the sciatic nerve) appeared devoid of INV at E17.5. Thus, as those nerves are very thin and mostly unmyelinated or poorly myelinated in the adult, the timing of vascularization may be different from what is observed in the sciatic nerve, or devoid of blood vessels as their caliber might allow oxygen diffusion from peri-nervous vascularization. One could thus wonder about the relationship between nerve type and INV development or absence of blood vessels.

Microvascular density is adapted to tissue oxygen and metabolic needs. During embryonic and post-natal development, nerves gain size and volume, due to myelination and production of the connective tissues. We hypothesized that this leads to increased blood vessel density, mainly based on potential formation of hypoxic zones inside the nerve during nerve growth driving angiogenesis. Interestingly, we found the opposite: the vascular tree stabilizes, angiogenesis is stopped, or remains very low, and as a consequence INV density decreases during post-natal development. This suggests that nerves reach their optimal blood vessel density around P10 similarly to the CNS in which angiogenesis continues until P10 in mice (*Harb et al., 2013*). At P5, there is still angiogenesis ongoing but at a much lower rate. Thus, myelination rate is faster than vascularization, leading to vascular density decrease and stabilization around P10, when both processes seem to end.

Even though SC and blood vessels are significant players in peripheral neuropathies pathogenesis and nerve repair mechanism (*Cattin et al., 2015*), no studies were available regarding their relationship during physiological development. In the developing PNS, the vascularization needs to be tightly controlled and we have shown that SC have a central role. To our knowledge, we here provide the first evidence that myelinating SC control the vascularization of the sciatic nerve during development. As the major phenomenon of the post-natal development of the PNS is myelination and that concomitantly the INV density is decreasing, we hypothesized that myelinating SC may be responsible for the blocking of angiogenesis. We were able to discriminate specific role of SC and myelination by genetically ablating either SC or myelin production. In both cases, we found that this ablation led to a hypervascularized nerve. Nevertheless, we observed that SC death provokes a more disrupted and immature INV, as compared to the phenotype of sciatic nerves only lacking myelin sheaths. Mukouyama and colleagues have shown that SC are required for arterial differentiation in the skin of mouse embryos (*Mukouyama et al., 2002*). This suggests that SC bodies provide physical and/or chemical cues that control blood vessel maturation. This is also what we observe as blood vessels of the INV failed to express 'barrier proteins' such as claudin-5 when myelin production was inhibited. Another study has suggested that SC-conditioned medium inhibits endothelial cell proliferation and migration in vitro (*Huang et al., 2000*), whereas another found that SC stimulates endothelial cell migration in vitro (*Ramos et al., 2015*). Since inhibition of myelination alone and total ablation of myelinating SC both led to hypervascularized sciatic nerves, this suggest that in vivo, SC may have in fact an anti-angiogenic effect on endothelial cells during post-natal development possibly by producing anti-angiogenic molecules. We cannot exclude the possibility that the hypervascularized sciatic nerve phenotype is due to an over-proliferation of promyelinating SC in *Krox20^{Cre/Fl}* mice. Yet, when SC are ablated in *Wnt1-Cre;Krox20^{GFP(DT)/+}* mice, INV at birth is also already more dense, suggesting that promyelinating SC could have an anti-angiogenic role. Thus, our results are in favor of a permissive environment provided by unmyelinating SC rather than a pro-angiogenic effect of over-numerous and proliferative SC. Despite the fact that *Krox20^{Cre/Fl}* mutants exhibit the same axon density, the possibility that axons induce angiogenesis to cope with lack of trophic support has to be considered. Nevertheless, during development, SCP (present around E13) and iSC (from E15 to P0) provide trophic support for neurons allowing their survival and normal nerve fasciculation (*Fledrich et al., 2019*). In *Krox20^{Cre/Fl}* mutant mice, those cells are present (SC blocked in immature stage and unmyelinating SC which do not express Krox20) and are expected to support the metabolic demand of axons, as normal axonal density was found (*Decker et al., 2006*).

Moreover, compact myelin sheaths are dense structures, composed of lipids (*Salzer, 2015*), and could have biophysical properties not favorable or less permissive to the migration and propagation of blood vessels. This environment could also block the diffusion of pro-angiogenic factors produced by other cells in the nerve such as vascular endothelial growth factor expressed by neurons (*Li et al., 2013*).

Altogether, our data demonstrate that different stages of SC development orchestrate the INV formation: around E16 when the immature SC proliferate, INV starts to develop. Between P0 and P10, when myelination is happening, INV density decreases but maturation occurs. This anti-angiogenic effect of the glial cells appears to be the opposite in the CNS. Interestingly, it has been shown that, oligodendrocytes promote angiogenesis and endothelial cell proliferation in the white matter (*Yuen et al., 2014*).

Another question raised by our findings is do SC and myelin have a role in the maintenance of the INV? Since *Wnt1-Cre;Krox20$^{GFP(DT)/+}$* and *Krox20$^{Cre/Fl}$* mice die soon after birth (around P5 and P20, respectively), we could not study further in time the role of SC and myelin regarding the maintenance of the INV. To gain insights into this and to know if myelin is not only necessary for INV onset, but also for its maintenance, futures studies could be done in mice models of demyelination at adult stage using PLPCre-ERT2; Krox20$^{Fl/Fl}$ animals. This could also provide useful information on the role of INV in degenerative diseases in which myelin is targeted, as INV maintenance could perhaps delay disease progression by maintaining nerve homeostasis.

Guidance molecules controlling angiogenesis in different systems during embryogenesis are reported (*Michaelis, 2014*) but the molecular control of INV formation is not well known. We chose to focus on Netrin-1, a protein known to play a role in migration of different cell types, including endothelial cell, during development (*Bradford et al., 2009*). The role of Netrin-1 regarding angiogenesis remains controversial, and seems to be dual, as it is in axon guidance, depending on the receptor it binds to *Yang et al., 2007*. It has been shown that Netrin-1 stimulates angiogenesis in vitro and also in vivo, in the retina (*Park et al., 2004*). In the developing sciatic nerve, Netrin-1 is expressed by the epi-perineurial cells with a concomitant expression of its receptor UNC5B by endothelial cells composing the INV. Despite the possible involvement of other factors, both independent loss of Netrin-1 or UNC5B function was responsible for INV formation defects, even though this was not sufficient to completely block nerve's vascularization. Indeed, in our study, Netrin-1 hypomorph provokes hypovascularized sciatic nerves at E17.5 and the blocking of UNC5B receptors at P0 leads also to a less vascularized sciatic nerve. This point out that the molecular couple Netrin-1/UNC5B is involved in INV establishment.

Since the formation of INV is finely regulated, molecularly and cellularly, by actors we have identified, the data presented here suggest a novel role for Netrin-1 and SC during the coordinated wiring of the nervous and vascular systems. Our study opens the exciting possibility that INV and SC/myelin could be linked not only during development but also during peripheral nerve diseases progression. Thus, INV implication in demyelinating diseases should not be overlooked.

# Materials and methods

**Key resources table**

| Reagent type (species) or resource | Designation | Source or reference | Identifiers | Additional information |
|---|---|---|---|---|
| Strain, strain background (*Mus musculus*; male/female) | *Krox20$^{cre}$* | Jackson laboratory | Egr2tm2(cre)Pch/J | B6D2 background |
| Strain, strain background (*Mus musculus*; male/female) | *Krox20$^{fl/+}$* | *Decker et al., 2006*; DOI: 10.1523/ JNEUROSCI.0716–06.2006 | | B6D2 background |
| Strain, strain background (*Mus musculus*; male/female) | *Krox20GFP(DT)* | *Vermeren et al., 2003* DOI: 10.1016/S0896-6273(02)01188–1 | | B6D2 background |
| Strain, strain background (*Mus musculus*; male/female) | Wnt1-Cre | Jackson laboratory | B6.Cg-$^{E2f1tg(Wnt1-cre)2Sor}$/J RRID:IMSR_JAX:022501 | B6D2 background |

*Continued on next page*

*Continued*

| Reagent type (species) or resource | Designation | Source or reference | Identifiers | Additional information |
|---|---|---|---|---|
| Strain, strain background (*Mus musculus*; male/female) | Cx-40-GFP | *Miquerol et al., 2004*; DOI:10.1016/j.cardiores.2004.03.007 | | CD1 background |
| Strain, strain background (*Mus musculus*; male/female) | Ng2-DsRed | Jackson laboratory | (TgCspg4-DsRed.T1)1Akik/J | C57/Bl6 background |
| Strain, strain background (*Mus musculus*; male/female) | *Ntn1^LacZ/+^* | *Serafini et al., 1996*; DOI: 10.1016/S0092-8674(00)81,795X | | CD1 background |
| Antibody | CD31 (rat monoclonal) | BD Pharmigen | Cat# 553370, RRID:AB_394816 | (1:400) |
| Antibody | CD31 (goat polyclonal) | R&D | Cat# AF3628, RRID:AB_2161028 | (1:400) |
| Antibody | Tuj-1 (mouse monoclonal IgG2a) | R&D | Cat# BAM1195, RRID:AB_356859 | (1:500) |
| Antibody | Tuj-1 (mouse monoclonal IgG2a) | Biolegend | Cat# 801213, RRID:AB_2728521 | (1:400) |
| Antibody | SMA-cy3 (mouse monoclonal) | Sigma | Cat# C6198, RRID:AB_476856 | (1:500) |
| Antibody | Neurofilament Heavy chain (chicken polyclonal) | Abcam | Cat# ab4680, RRID:AB_304560 | (1:1000) |
| Antibody | Ki67 (rabbit polyclonal) | Abcam | Cat# ab15580, RRID:AB_443209 | (1:500) |
| Antibody | Sox10 (Mouse monoclonal IgG2a) | Proteintech | Cat# 66786–1-Ig, RRID:AB_2882131 | (1:200) |
| Antibody | S100β (rabbit polyclonal) | Proteintech | Cat# 15146–1-AP, RRID:AB_2254244 | (1:400) |
| Antibody | β-Galactosidase (chicken polyclonal) | Abcam | Cat# ab9361, RRID:AB_307210 | (1:500) |
| Antibody | Goat anti-Rat IgG (H + L) Alexa Fluor 555 (goat polyclonal) | Thermo Fisher | Cat# A-21434, RRID:AB_2535855 | (1:400) |
| Antibody | Goat anti-Chicken IgG (H + L) Alexa Fluor 647 (goat polyclonal) | Thermo Fisher | Cat# A32933, RRID:AB_2762845 | (1:400) |
| Antibody | Donkey anti-Rabbit IgG (H + L) Alexa Fluor 488 (donkey polyclonal) | Thermo Fisher | Cat# A-21206, RRID:AB_2535792 | (1:400) |
| Antibody | Donkey anti-Goat IgG (H + L) Alexa Fluor 555 (donkey polyclonal) | Thermo Fisher | Cat# A-21432, RRID:AB_2535853 | (1:400) |
| Antibody | Rat UNC5H2/UNC5B Antibody | Bio-Techne | AF1006 | (dose = 4 µg/g of body weight) IF (1:200) |
| Commercial assay or kit | RNAscope Multiplex Fluorescent V2 Assay | ACD | 323100 | |
| Sequence-based reagent | RNAscope Probe - Mm-Ntn1 | ACD | 407621 | |
| Sequence-based reagent | RNAscope Probe - Mm-Fap-C3 | ACD | 423881-C3 | |
| Sequence-based reagent | RNAscope Probe - Mm-Pecam1-C3 | ACD | 316721-C3 | |

*Continued on next page*

*Continued*

| Reagent type (species) or resource | Designation | Source or reference | Identifiers | Additional information |
|---|---|---|---|---|
| Sequence-based reagent | RNAscope Probe - Mm-Unc5b-No-XHs | ACD | 482481 | |
| Sequence-based reagent | PrimePCR Template for SYBR Green Assay: Pecam1, Mouse | Bio-Rad | qMmuCID0005317 | |
| Sequence-based reagent | PrimePCR Template for SYBR Green Assay: Dcc, Mouse | Bio-Rad | qMmuCED0051027 | |
| Sequence-based reagent | PrimePCR Template for SYBR Green Assay: neo1, Mouse | Bio-Rad | qMmuCID0011752 | |
| Sequence-based reagent | PrimePCR Template for SYBR Green Assay: unc5b, Mouse | Bio-Rad | qMmuCID0016421 | |
| Commercial assay or kit | NucleoSpin RNA Plus XS, Micro kit for RNA purification | Macherey-Nagel | REF 740990.50 | |

## Animals

C57BL/6 (Janvier labs, France) mice were used for this study. *Cx40-GFP* (*Miquerol et al., 2004*), *Ng2-DsRed* (Tg(Cspg4-DsRed.T1)1Akik/J; Jackson laboratory) mice were previously described and genotyped with epifluorescence microscope. *Krox20*$^{GFP(DT)}$ (*Vermeren et al., 2003*), *Krox-20-Cre* (Egr2tm2(cre)Pch/J; Jackson laboratory), *Wnt1-Cre* (129S4.Cg-E2f1Tg(Wnt1-cre)2Sor/J; Jackson laboratory), *Krox20*$^{fl/+}$ (*Taillebourg et al., 2002*), and *Ntn1*$^{LacZ/+}$ (*Serafini et al., 1996* ) mice were previously described and genotyped by PCR. For embryonic stages, the day of the vaginal plug was counted as E0.5.

Experiments and techniques reported here complied with the ethical rules of the French agency for animal experimentation.

## Injection of anti-UNC5B blocking antibody

Swiss WT pups were injected intra-peritoneally at P0 with 40 µL of anti-UNC5B blocking antibodies (Bio-Techne) (*König et al., 2012*; *Tadagavadi et al., 2010*) diluted with physiological serum (0.9% NaCl) at a final dose of 4 µg/g of body weight and control littermates were injected with 40 µL of physiological serum; 24 hr later, pups were sacrificed and sciatic nerves were dissected followed by whole-mount immunofluorescent staining.

## Immunofluorescent staining

After euthanasia, sciatic nerves were dissected and fixed in a 4% PFA solution during 30 min at room temperature. The whole-mounts were incubated in a TNBT solution composed of Tris HCl pH 7.4/ NaCl 5 M/0.5% blocking reagent (Perkin)/0.5% Triton X-100, overnight at 4°C. Primary antibodies were diluted in the same solution, overnight at 4°C. After washes with TNT solution (Tris pH 7.4/ NaCl 5 M/0.05% Triton X-100), nerves were incubated with secondary antibodies, diluted in TNBT solution, during 3 hr at room temperature. For cryostat sections: sciatic nerves, immediately after being dissected, were embedded in OCT media and snap-frozen in liquid nitrogen. Cryostat sections (14 µm) were fixed using ice-cold 100% methanol during 8 min then incubated in a blocking solution composed of 0.25% Triton X-100/10% fetal bovine serum/PBS during 30 min at room temperature. Primary antibodies were diluted in the same solution and sections were incubated overnight at 4°C. After PBS washes, secondary antibodies were diluted in a 0.1% Triton X-100/1% FBS/PBS solution and sections were incubated during 2 hr at room temperature. All the antibodies used in this study together with the information regarding their use are listed in the key resources table.

## Tissue clearing

For embryonic sciatic nerves and paws, we used the iDisco+ clearing method (*Renier et al., 2014*). Briefly, after immunofluorescent staining and inclusion in 1% agarose blocks (to facilitate imaging), samples were first dehydrated with methanol/H$_2$O series (20%, 40%, 60%, 80%, and 100%; 1 hr each) at room temperature and then incubated in 66% dichloromethane (DCM, Sigma Aldrich)/33%

methanol during 3 hr. This was followed by an incubation with 100% DCM 15 min, twice. Samples were then cleared with dibenzyl ether (DBE, Sigma Aldrich), overnight.

## Fluorescent in situ hybridization

RNAscope Multiplex Fluorescent V2 Assay (ACD) kit was used to detect mRNA on sciatic nerve. Briefly, after dissection, sciatic nerves were fixed with 4% PFA solution overnight, at 4°C and after three washes with PBS, they were cryoprotected with a solution of sucrose 30% overnight at 4°C. The samples were then embedded in OCT and conserved at –80°C. Longitudinal cryo-sections of 14 μm were done and RNAscope Assay was carried out according to manufacturer's protocol. The probes used are detailed in the key resources table.

## Western blot

Sciatic nerves of WT mice were lysed in a protein extraction buffer (50 mM Tris, 150 mM NaCl, 5 mM EDTA, 1% Triton-X100, 1% SDS, 1% protease inhibitor) and homogenized with sterile tungsten beads using the homogenizer (Qiagen) for 45 s at 30 beats per second. The supernatants obtained after centrifugation (14,000 rpm for 15 min at 4°C) were used as samples. The extracted proteins were assayed with the Bio-Rad kit (Hercules, CA) by spectrophotometry using BSA as a standard. Equal quantities of proteins (15 μg) were denatured by heating in Laemmli buffer containing β-mercaptoethanol, then separated in a 10% SDS-polyacrylamide gel. The proteins were then transferred to a Hybond-C nitrocellulose membrane (Amersham Biosciences). The non-specific binding sites of the membrane were blocked in Tris buffer containing 0.4% Tween (TBS-T) supplemented with 5% milk powder, for 3 hr at room temperature. The membrane was then incubated in TBS-T supplemented with 1% milk powder, mixed with rabbit polyclonal antibodies to Krox20 (1/2000, Covance, Berkeley, CA) overnight at 4°C. After washing, it was incubated with an anti-rabbit antibody conjugated with HRP peroxidase (1/5000) (Amersham Biosciences), for 1 hr at room temperature. Actin was labelled with mouse monoclonal anti-β-actin primary antibodies (1/5000) followed by HRP-conjugated anti-mouse secondary antibodies (1/5000). The revelation was made by chemiluminescence with the Supersignal West Femto kit (Thermoscientific). The signals obtained were acquired and digitized on the Chemicapt automaton.

## Imaging and images processing

The images of nerves sections were taken using Zeiss Axiozoom apotome (and associated Zen software) and Leica confocal microscope. Maximum projections of the acquired stacks were obtained with Fiji. The cleared tissues images were acquired using a light-sheet microscope and Inspector pro software (Lavision biotec). The 3D reconstruction of nerves and limbs were visualized and analyzed with Imaris software (Bitplane).

## Quantifications

### Number of angiogenic sprouts

The number of angiogenic sprouts from sciatic nerves (whole mount), at different developmental stages, was counted manually using a confocal microscope (Leica, ×40). An endothelial cell was considered to be a tip cell if it displayed at least two distinct filopodias. This number per nerve was then normalized to a surface of 1 $mm^2$.

### Ratio of Cx-40+ and NG2+ blood vessel coverage

For each stage, three to four whole mounts of dissected sciatic nerves were imaged using axiozoom (Zeiss) and after thresholding of the images using Fiji software, a ratio of NG2+ (or Cx-40+) raw integrated signal density over CD31 raw integrated signal density was calculated.

### Quantification of the nerve area

In order to detect the sciatic nerve on whole-mount, sections, and cleared limbs, the axonal markers NF or Tuj-1 were used. During images processing, the signal of this staining was used to delimit the nerve area in $μm^2$ using Fiji (methods *a* and *b*; *Figure 7A and C*). From cleared limbs, the sciatic nerve was first localized in 3D and we always analyzed the same portion of the nerve. This portion starts just after the node formed by the three branches coming from the DRG and stops just before the

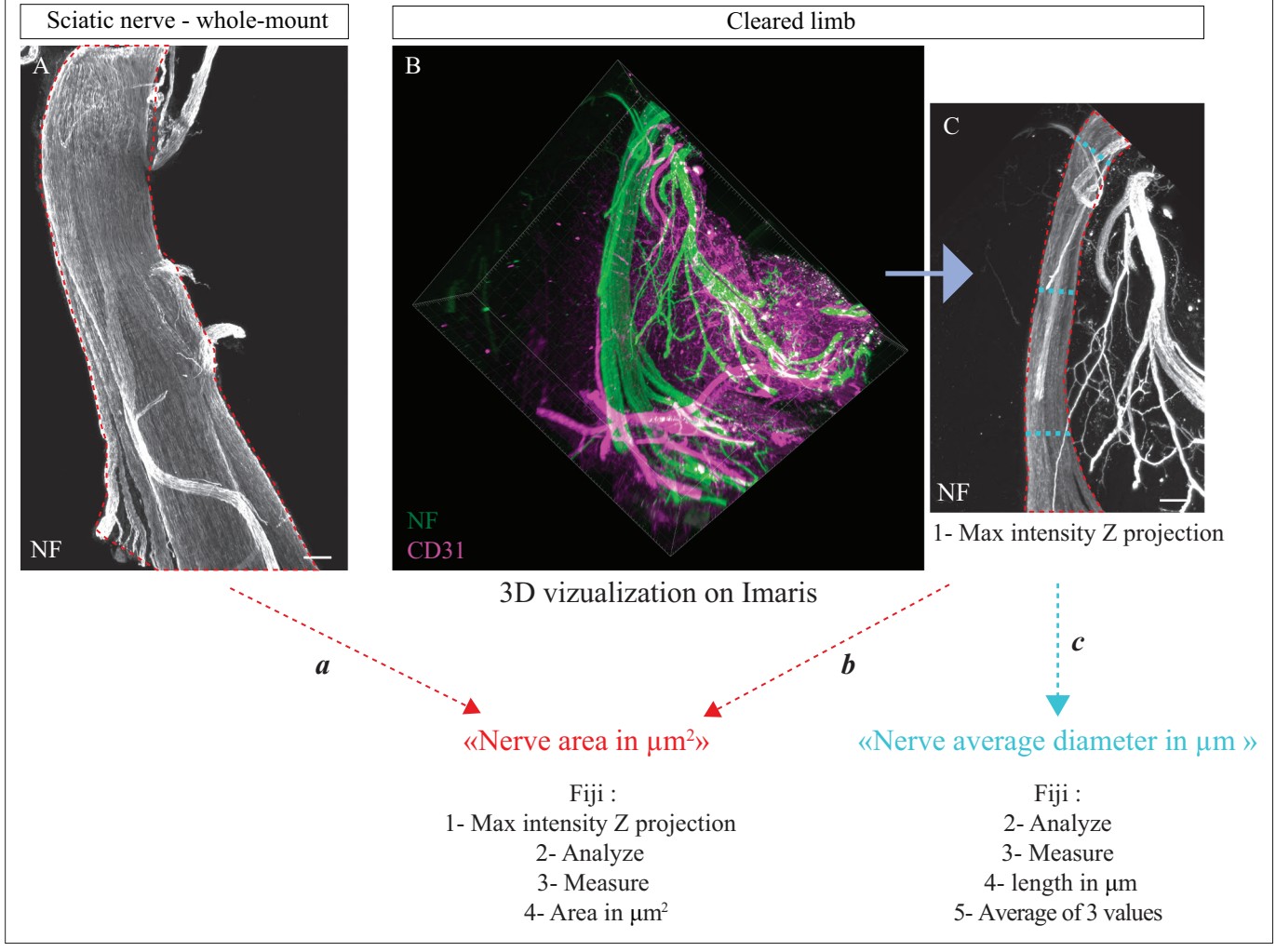

**Figure 7.** Methodology used to quantify nerve area. (**A**) Whole-mount immunofluorescent staining of a sciatic nerve showing axons in white (neurofilament [NF]) at P0. Red dotted lines represent nerve area. (**B**) Snapshot of a 3D visualization of a cleared limb at E17.5, axons are marked in green (NF) and blood vessels in magenta (CD31). (**C**) Maximum intensity Z projection of sciatic nerve portion from cleared limb images, axons are in white (NF). Red dotted lines represent nerve area and blue dotted lines represent diameter measurements. Scale bars are 100 μm.

first distal branches. For analysis of nerve thickness in *Ntn1^{LacZ/LacZ}* and *Ntn1^{+/+}* at E16, method 'c' was used with Fiji on the same nerve portion. Overall, method *a* was used for *Figure 4F*, *Figure 4—figure supplement 1B'*, *Figure 4—figure supplement 2B'*, *Figure 5D*, *Figure 6E*, method *b* for *Figure 6H* and method *c* for *Figure 6—figure supplement 1F*.

## INV density from sciatic nerves' sections

The density of intra-nervous blood vessels on sciatic nerve cross sections, at different developmental stages, was quantified using the same semi-automatic method on Fiji for all the sections. After thresholding the images and creating a mask for the channel representing the blood vessels' staining, the total area contained in the section was quantified. This value was reported to the area of the section.

## 3D blood vessel length and number of branch points

For the quantification of the length and number of branch points of the vasculature, it was done using 'Surface' and 'Filament' tools of Imaris software.

For *Ntn1^{+/+}* and *Ntn1^{LacZ/LacZ}* embryos, quantification of the nerve area (NF staining) and intra-nervous blood vessel density (CD31 staining) was made using Fiji software.

## Quantification of FISH

From sciatic nerve longitudinal sections, the epi-perineural area was considered as the most external cell layer containing *Fap+* cells. The central part was considered as endonerve. After image thresholding using Fiji software, the number of *Ntn1+* dots was quantified and divided by the area of the corresponding layer (epi-perinerve and endonerve). For *unc5b*, blood vessels were delimited using *cd31* signal and after thresholding, the number of *unc5b+* dots was quantified and divided by the area of the blood vessels or by the area of the rest of the nerve.

## Quantification of Netrin1 and UNC5B immunofluorescent staining

After image thresholding using Fiji software, ratio of raw integrated signal densities of Netrin-1+/FAP+ and UNC5B+/CD31+ were calculated.

## Purification of endothelial cells and qPCR

Five tubes, each containing 10 sciatic nerves from P2 WT mice, were used. Sciatic nerves were digested at 37°C with enzymes from *Miltenyi* 'brain dissociation' kit. Then, positive selection of CD31+ cells was done by magnetic separation, following the protocol of *Miltenyi biotec.* The cells were counted with 'Countess' machine (*Thermo Fisher*): around $1 \times 10^4$ cells per tube was obtained. Total mRNA was extracted from those cells with the kit 'RNA plus XS' from *Macherey-Nagel.* qPCR was performed with *Bio-Rad Prime PCR* validated primers at 60°C. The normalized expression relative to β-actin and GAPDH was obtained with ΔΔCq method: ΔCq Expression = $2^{-\Delta Cq}$ with ΔCq = Cq (target gene) – Cq (reference) and Cq (reference) $=\sqrt{Cq\,(Bactin)\,xCq\,(GAPDH)}$ .

## Statistical analysis

Data shown are expressed as means ± SEM. Graphs and statistical analysis were performed using GraphPad Prism software, with p-value <0.05 considered as significant. The different statistical tests used were specified in the figures' legends. Shapiro-Wilk normality test was used to assess if the values come from a Gaussian distribution.

# Acknowledgements

We acknowledge Philippe Mailly for the help in image analyses. We would also like to thank the Fondation Bettencourt Schueller. This work was supported by *la Ligue contre le cancer*, *La fondation College de France*, and Labex Memolife.

# Additional information

## Funding

| Funder | Grant reference number | Author |
|---|---|---|
| Ligue Contre le Cancer | | Sonia Taïb |
| Fondation du Collège de France | | Sonia Taïb<br>Isabelle Brunet |
| LabEx Memolife | | Sonia Taïb |

The funders had no role in study design, data collection and interpretation, or the decision to submit the work for publication.

## Author contributions

Sonia Taïb, Conceptualization, Formal analysis, Investigation, Methodology, Validation, Visualization, Writing - original draft; Noël Lamandé, Sabrina Martin, Investigation; Fanny Coulpier, Data curation, Resources; Piotr Topilko, Resources, Writing - review and editing; Isabelle Brunet, Conceptualization, Funding acquisition, Methodology, Project administration, Supervision, Validation, Writing - original draft, Writing - review and editing

## Author ORCIDs

Sonia Taïb [ID] http://orcid.org/0000-0002-9981-5204
Piotr Topilko [ID] http://orcid.org/0000-0001-7381-6770
Isabelle Brunet [ID] http://orcid.org/0000-0002-5490-2937

## Decision letter and Author response

Decision letter https://doi.org/10.7554/eLife.64773.sa1
Author response https://doi.org/10.7554/eLife.64773.sa2

## Additional files

### Supplementary files

• Transparent reporting form

### Data availability

All data generated or analysed during this study are included in the manuscript and supporting files. Source data files have been provided for all figures.

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
