## [Editor Report]

This manuscript focuses on the cellular and molecular mechanisms underlying intra-nervous vascularisation of peripheral nerves during embryogenesis and early postnatal development. While the general molecular principles of angiogenesis and peripheral nerve development have been described, how these two processes are coordinated to form the intranervous vascular system is virtually unknown. Using mouse genetic models, the authors show that Schwann cells regulate vascularization of the sciatic nerve and are required for a decrease in vascular density postnatally.

---

## [Decision Letter]

**Decision letter after peer review:**

Thank you for submitting your article "Myelinating Schwann cells and Netrin-1 control intra-nervous vascularization of the developing mouse sciatic nerve" for consideration by *eLife*. Your article has been reviewed by 3 peer reviewers, and the evaluation has been overseen by Karina Yaniv as Reviewing Editor and Anna Akhmanova as the Senior Editor. The reviewers have opted to remain anonymous.

The reviewers have discussed the reviews with one another and the Reviewing Editor has drafted this decision to help you prepare a revised submission.

Summary:

This manuscript focuses on the cellular and molecular mechanisms underlying intra-nervous vascularisation of peripheral nerves during embryogenesis and early postnatal development.

While the general molecular principles of angiogenesis and peripheral nerve development have been described, how these two processes are coordinated to form the intranervous vascular system (INV) is virtually unknown. Using mouse genetic models, the authors find that Schwann Cells (SCs) regulate vascularization of the sciatic nerve and are required for a decrease in vascular density postnatally. They initially use immunofluorescence to describe the timeline of INV development and then test the relationship between nerve vascularisation and myelination by examining INV development following Schwann cell ablation. In addition, the authors show that in Netrin1 mutant mice, peripheral nerve vascularisation is reduced, suggesting Netrin1 as a molecular regulator of this process.

Overall, the reviewers find this work interesting and novel. While they agree that the technical level of the work is adequate and the manuscript is well written, they also feel that some of the data appear a bit preliminary and that more detailed and deeper analyses would allow a firmer interpretation of many of the experiments.

The reviewers have discussed these issues with each other after submitting their reviews, and they agreed that the following points will be essential in order for the manuscript to be considered for publication:

Essential revisions:

1. The effects of the Krox20 knockout in Schwann cells: In order to better describe the role of SCs in sciatic nerve vascularization, it would be important to understand whether deletion of Krox20 in promyelinating Schwan cells does not alter any other property in the nerve beside the lack of myelination. As the authors show, Krox20 mutant Schwann cell precursors are highly proliferative and remain undifferentiated. Could the effects on INV be caused by a signal originating from these abnormal cells, rather than by the absence of myelin per se? By ascribing most of the phenotype to the absence of myelin in Krox20, the authors might be over-interpreting their data. Similarly, as SCs are known to provide metabolic support to axons, the possibility exists that when this support is not provided, axons respond by inducing angiogenesis to cope with their metabolic demand.

2. The role of Netrin1: It would be important to understand which cells express netrin-1, what is the source of Netrin-1 that regulates vascularization (neurons, Schwann cells, pericytes?) and whether netrin-1 acts directly on endothelial cells? In addition, any extra data the authors can show or discuss on the netrin receptors expressed by endothelial cells in the growing vessels, would be very interesting for this paper.

3. The authors should rule out involvement of Netrin1 in the development of peripheral nerves.

4. Additional markers need to be analyzed for the different cell types assessed. For example, endothelial cells are only identified using CD31, despite many other markers being available.

5. Quantifications need to be provided for all experiments. Additionally, the authors should provide clear description of the number of animals analysed in each experiment. For some experiments, it is not clear whether the n is the number of limbs or individual animals. For other experiments, the number of animals in which the observations are made is not reported at all (Figure 1 etc). Also, additional controls/normalisations regarding the size of the nerves are necessary. The nerve looks smaller in Krox20 mutants…

[Editors' note: further revisions were suggested prior to acceptance, as described below.]

Thank you for resubmitting your work entitled "Myelinating Schwann cells and Netrin-1 control intra-nervous vascularization of the developing mouse sciatic nerve" for further consideration by *eLife*. Your revised article has been evaluated by Anna Akhmanova (Senior Editor), Karina Yaniv as Reviewing Editor and the original reviewers.

While the reviewers agree that some of their concerns have been satisfactorily solved, there are remaining issues that need to be addressed in order for the manuscript to be considered for publication. The reviewers feel that the potential of this study is not fully developed with many opportunities and controls missed. The reviewers emphasize that they do not expect the authors to severely expand the scope of the study, but rather to be as thorough as expected. In light of their comments, addition of the experimental data requested below, will be absolutely essential for further consideration of this manuscript.

1. In general, the reviewers find the provided quantifications and its statistical power to be convincing. A remaining concern in this regard relates to the size measurements of the sciatic nerve. For instance, the sciatic nerve in Figure 6 and its S1 is very different between mutant and controls in terms of its branching pattern, thickness etc. This rises some doubts about whether the measurement are done consistently in all animals. The authors need to provide additional information and examples on how the nerve size was measured.

2. An additional major problem is that some of the reviewers' concerns have been addressed through the rebuttal letter without any significant or explicit edits in the manuscript. For example, the comment #1 about the specificity of the effects of the Krox20 mutation on myelination gets 5 paragraphs and many references in the rebuttal letter, but only a Results section mention that the sciatic nerves "from Krox20 Cre/Flmice are devoid of myelin" (line 218). Generally, in the rebuttal, the authors use unpublished observations, and references to on-line databases but these are not validated. This should be fully amended in the revised version.

3. The identity of netrin1 expressing cells is not convincing and needs further work. The authors added a new result, showing that at E17 (onset of vascularization) the netrin1 signal appears close to the developing vasculature, as opposed to E16.5, where netrin signal is dispersed (supp. Figure 6A, B). They suggest that at P1 Netrin1 is expressed by the epineurial and perineurial cells and proliferating fibroblast like cells, based on a recent scRNA-seq analysis from Gerber et al., 2021. However, the authors didn't provide any validation to this using relevant markers to show the co-expression with these cells. The qPCR quantification of netrin1 receptor expression also falls short of a developmental paper. What is missing is a demonstration of mRNA expression by in situ, for example using RNAscope.

4. The authors did a good job demonstrating the development of the sciatic nerve vascularization and that the elimination of the Schwann cells/myelination disrupts the regulation of the angiogenesis, however, it is still not clear whether this is a direct effect, or secondary effect due to affected neurons. Netrin1 is required for the normal development of peripheral nerves. Poliak et al. (PMID: 26633881) show this explicitly. Netrin1 mutation (same one as used here) affects the guidance of motor axons within peripheral nerves, including the sciatic nerve. The authors should extend the analysis to additional nerves (e.g., the diaphragm, which is a classic peripheral nerve, or purely sensory/cutaneous nerves) to convincingly argue that the phenotypes are generalizable to different peripheral nerves, as previously suggested by the reviewers.

5. The expression of the suggested Netrin1 receptors (unc5b) by the endothelial cells of the sciatic nerve – the data provided by the authors are not sufficient to suggest that Netrin1 regulates angiogenesis through this receptor. Although this might require the analysis of another transgenic line, or perhaps isolation and culture of ECs from sciatic nerves, this part would be a clear prove of the direct role of Netrin1 in the sciatic vasculature.

6. Throughout Figure 1 the authors use the "tip cell" term to refer to angiogenic sprouts. However, endothelial tip cells are characterized by the presence of specific markers that the authors do not provide in their characterization. Hence the definition and corresponding quantification should be changed to either "branching points", "end points" or angiogenic sprouts. All these are standardized measurements offered by different image processing tools.

---

## [Author Response]

Essential revisions:1. The effects of the Krox20 knockout in Schwann cells: In order to better describe the role of SCs in sciatic nerve vascularization, it would be important to understand whether deletion of Krox20 in promyelinating Schwan cells does not alter any other property in the nerve beside the lack of myelination. As the authors show, Krox20 mutant Schwann cell precursors are highly proliferative and remain undifferentiated. Could the effects on INV be caused by a signal originating from these abnormal cells, rather than by the absence of myelin per se? By ascribing most of the phenotype to the absence of myelin in Krox20, the authors might be over-interpreting their data. Similarly, as SCs are known to provide metabolic support to axons, the possibility exists that when this support is not provided, axons respond by inducing angiogenesis to cope with their metabolic demand.

Beside the lack of myelination, Krox20 deletion in promyelinating Schwan cells does not alter any other property in the nerve, as described in Decker et al. 2006. In this original paper characterizing and describing the phenotype of *Krox20^cre/Fl^* mutant mice, the authors showed, using electron microscopy, a complete absence of the myelin sheath, despite a normal axonal density (89 ± 2 and 88 ± 7 axonal fibers per 100 μm2 in wild-type and *Krox20^Cre/flox^* mice, respectively) in the postnatal sciatic nerve (P28) (Decker et al. 2006).

In the sciatic nerve, transcription of *Krox20* gene is activated at E15 but the protein is detectable shortly after birth (see Figure 5-figure supplement 1). Consistent with these results, general organization of the mutant sciatic nerve, including proportion of iSC are not affected between E15.5 and E18.5. Finally, there is no significant differences from microarray comparison of *Krox20^Cre/flox^* mutant versus control nerves at E18.5 (unpublished data from P.Topilko). In conclusion, development of the sciatic nerve of *Krox20^Cre/flox^* mice is unaffected until birth (whereas the INV onset is at E16,5) and thus early angiogenesis effects leading to INV formation should not be affected in the mutant.

After birth, SC progressively stop dividing to initiate myelination process. In *Krox20^Cre/flox^* mice, the lack of myelin is intimately correlated with hyper proliferation of SC in order to cover all axons. To our knowledge, there is no mutant where myelination is affected while SC number remains unchanged. Thus, to not over-interpret the data in our study, we added a short paragraph in the discussion addressing possibility that the hypervascularization phenotype of the mutant sciatic nerve might be due to an over proliferation of promyelinating SC. Yet, genetic SC ablation in *Wnt1-*cre;*Krox20*^GFP(DT)/+^ mice promote increased INV density at birth, suggesting that promyelinating SC could have an anti-angiogenic role. Moreover, INV density decreases during post-natal stages, while myelin is progressively produced (showed in figure 3) and this observation also supports the hypothesis that myelinated SC negatively control angiogenesis. Thus, our results are in favor of unmyelinated SC providing a permissive environment to endothelial cells to vascularize the nerve tissue rather than a pro-angiogenic effect of over numerous and proliferative SC.

Despite the fact that postnatal mutants exhibit the same axon density, the possibility that axons induce angiogenesis to cope with lack of trophic support was raised. Nevertheless, we know that during development, Schwann cell precursors (SCP) (present around E13) and immature SC (from E15 to P0) provide trophic support for neurons allowing their survival and normal nerve fasciculation (Fledrich et al. 2019). In *Krox20^cre/Fl^* mutant mice, those cells are present (SC blocked in immature stage and unmyelinating SC which do not express Krox20) and should support the metabolic demand of axons (as normal axonal density was found (Decker et al. 2006)).

2. The role of Netrin1: It would be important to understand which cells express netrin-1, what is the source of Netrin-1 that regulates vascularization (neurons, Schwann cells, pericytes?) and whether netrin-1 acts directly on endothelial cells? In addition, any extra data the authors can show or discuss on the netrin receptors expressed by endothelial cells in the growing vessels, would be very interesting for this paper.

As Netrin-1 is a secreted protein, we first tried to label it directly by immunofluorescent staining but it did not give satisfying results (and protein secretion makes it difficult to accurately identify producing cells). Thus, we chose to use an anti-β-galactosidase antibody to indirectly visualize Netrin-1 expression and sources in Ntn1^LacZ/+^ reporter mice. At E16, before INV onset, we found no β-galactosidase expression near the sciatic nerve (figure 6 S1 A). At E17,5, when the INV onset has started, β-galactosidase seems to be expressed by the nerve and particularly at the periphery, at the level of the epinerve/perinerve (figure 6 S1 B). Furthermore, in a recent paper, Gerber and colleagues have published the single cell RNA sequencing analysis of the different cell types constituting sciatic nerves, during perinatal development (Gerber et al. 2021). According to the atlas on this study (https://snat.ethz.ch/), *Ntn1* is in fact expressed at P1 by epineurial and perineurial cells and proliferating fibroblast like cells. This data corroborates our results as this cell type profile fits the in situ 3D observation we made (figure 6 S1 B).

Concerning the presence and/or the identification of Netrin-1 receptors, we purified endothelial cells from sciatic nerves at P2 using magnetic CD31- beads and performed RNA extraction followed by qPCR. We found that *unc5b* is highly expressed by endothelial cells, whereas *neogenin1* is weakly expressed and *dcc* is not expressed (figure 6B). This result is supported by the data of the published atlas: *dcc* is not expressed by endothelial cells of sciatic nerve at P1, *neogenin1* weakly expressed and *unc5b* highly expressed (Gerber et al. 2021).

Altogether, those data support the hypothesis of a direct effect of Netrin1 on sciatic nerve endothelial cells, as both the ligand and receptors are present at the right time and place to allow a direct angiogenic response.

Those data were included and discussed in the main text page 12 (Netrin1 localization and Netrin1 receptors expression).

3. The authors should rule out involvement of Netrin1 in the development of peripheral nerves.

In order to only study the role of Netrin-1 regarding the INV development, we chose to use the *Ntn1^LacZ^* hypomorph model rather than a complete null. Indeed, in the *Ntn1^LacZ^* hypomorph, some of the wild-type transcripts are present, presumably by splicing over the inserted sequences, resulting in milder developmental defects (Serafini et al. 1996). In fact, whereas Ntn1 null animals commissural axons rarely cross the midline, resulting in strong phenotype defects at late embryonic stages, Ntn1 hypomorphs retain many axons with normal trajectories. Thus, low levels of Ntn1 can account for persistent attraction to the midline in hypomorphs (Yung, Nishitani, and Goodrich 2015).

In *Ntn1^LacZ/LacZ^* embryos at E10.5, lack of strong ventral attraction by netrin 1/Dcc signalling could enable CNS-derived interneurons to grow towards weaker attractive signals in the periphery, leading to CNS exit. But at older age (E12,5), centrally located neurons project axons into the periphery of *Ntn1^LacZ/LacZ^* embryos and no significant over numerous spinal interneuronal axons was found in the dorsal root ganglia or at a different axial level (Laumonnerie et al. 2014).

In addition, the majority of published papers are about the role of Netrin-1 signaling in peripheral nerve regeneration (Dun and Parkinson 2017). In fact, only few data concerning the role of the axonal cue Netrin-1 in the development of peripheral nerves during late embryogenesis are available in the literature, but suggesting a minor impact of Netrin-1 (Bin et al. 2015; Laumonnerie et al. 2014).

Nevertheless, to rule out a potential involvement of Netrin-1 specifically during sciatic nerve development and at the onset of vascularization, we quantified and compared the sciatic nerve diameter at E16 of *Ntn1^LacZ/LacZ^* and *Ntn1^+/+^* control embryos. We found no difference between the two groups (figure 6 S1 C) indicating that Netrin-1 does not affect nerve formation in the *Ntn1^LacZ/LacZ^* mutant. This could be due to either (1) Netrin-1 not being involved in sciatic nerve formation or (2) implicated at early guidance steps, but already compensated by other molecular cues guiding pioneer axons allowing migration and fasciculation of other axons to form primitive nerve properly in *Ntn1^LacZ/LacZ^ embryos*.

4. Additional markers need to be analyzed for the different cell types assessed. For example, endothelial cells are only identified using CD31, despite many other markers being available.

In our study, we aim to describe and quantify intra-nervous vasculature during development. So we had to choose the best marker to homogeneously stain blood vessels at all stages and in the mutant mice. Indeed, one of the main goals of our study was to be able to compare each stage and samples with quantification of exact same parameters. We also wanted to compare the spatio-temporal organization of INV that require identical markers. In order to do so, we chose CD31 as it is the most specific marker of endothelial cells, and it is expressed constitutively on the surface of these cells at all stages throughout the development. It was also the optimized marker to reconstruct in 3D and accurately quantify the vascular network and to visualize endothelial tip cells. Moreover, this marker is classically used in studies of the nerve’s vascular system. Thus, this choice allows a robust comparison with existing data from the literature.

In addition, we aimed to visualize SMA, NG2 and Cx40 to describe the maturation steps of the INV (arterial differentiation and mural cell recruitment), as it was not known (Figure 2). In the scope of another study, it would be interesting to study further the molecular identity of endothelial cells at specific stages or their potential heterogeneity. Again, our point here was not to extensively describe the INV molecular identity but rather its development (and maturation) and the cells/molecular cues involved in this phenomenon.

As for the SC, we used Sox10 as all SC expresses it at all stages, so it was convenient to mark SC in both the mutant and control mice. No additional markers were required because those mice were already described.

5. Quantifications need to be provided for all experiments. Additionally, the authors should provide clear description of the number of animals analysed in each experiment. For some experiments, it is not clear whether the n is the number of limbs or individual animals. For other experiments, the number of animals in which the observations are made is not reported at all (Figure 1 etc). Also, additional controls/normalisations regarding the size of the nerves are necessary. The nerve looks smaller in Krox20 mutants…

We fully agree with the reviewers that more quantitative data were needed, thus we added new quantifications in all figures (except Figure 3 with already quantified data) and we modified the text accordingly. We also added information regarding the number of animals in each figure legends. Unless otherwise stated, one sciatic nerve per animal is used for quantification.

The nerve size was used to normalize the INV in our manuscript as this is generally the case to describe and compare vascularization of organs (as vascularization rate is directly comparable without size/developmental consideration in the literature). To quantify the nerve size, an axonal staining (neurofilament or Tuj-1) was used and with fiji or imaris, we did a Z projection and delineated the nerve to obtain a surface in μm^2^.

For the quantification of the INV, when the nerve size was not different, no additional controls/normalizations were needed. However, when the nerve size was different, we considered the “vascular rate” (blood vessel length over nerve area) as a pertinent data to compare the level of vascularization. This for 3 reasons: (1) We believe it makes more sense physiologically to talk about the “vascular rate” to better appreciate the level of vascularization of a tissue. (2) This method is often used in the literature. (3) We searched for other methods but other parameters, notably the diameter, can vary more than the size. In particular, in *Wnt1-*cre;*Krox20*^GFP(DT)/+^ mice, the nerve is indeed smaller at P5 as compared to the control, that is why we normalized the blood vessel length and number of branch points with the nerve area to obtain the vascular rate. Additionally, we also analyzed those mice at P0, when the nerve size is the same (Figure 4 S2).

References

Bin, Jenea M., Dong Han, Karen Lai Wing Sun, Louis Philippe Croteau, Emilie Dumontier, Jean Francois Cloutier, Artur Kania, and Timothy E. Kennedy. 2015. “Complete Loss of Netrin-1 Results in Embryonic Lethality and Severe Axon Guidance Defects without Increased Neural Cell Death.” *Cell Reports* 12(7):1099–1106.

Decker, Laurence, Carole Desmarquet-Trin-Dinh, Emmanuel Taillebourg, Julien Ghislain, Jean Michel Vallat, and Patrick Charnay. 2006. “Peripheral Myelin Maintenance Is a Dynamic Process Requiring Constant Krox20 Expression.” *Journal of Neuroscience* 26(38):9771–79.

Dun, Xin-peng and David B. Parkinson. 2017. “Role of Netrin-1 Signaling in Nerve Regeneration.” *International Journal of Molecular Sciences* (18, 491):1–22.

Fledrich, Robert, Theresa Kungl, Klaus Armin Nave, and Ruth M. Stassart. 2019. “Axo-Glial Interdependence in Peripheral Nerve Development.” *Development (Cambridge, England)* 146(21):1–12.

Gerber, Daniel, Jorge A. Pereira, Joanne Gerber, Ge Tan, Slavica Dimitrieva, Ueli Suter, and Emilio Ya. 2021. “Transcriptional Profiling of Mouse Peripheral Nerves to the Single-Cell Level to Build a Sciatic Nerve ATlas ( SNAT ).” 1–28.

Laumonnerie, Christophe, Ronan V. Da Silva, Artur Kania, and Sara I. Wilson. 2014. “Netrin 1 and Dcc Signalling Are Required for Confinement of Central Axons within the Central Nervous System.” *Development (Cambridge)* 141(3):594–603.

Serafini, Tito, Sophia A. Colamarino, E. David Leonardo, Hao Wang, Rosa Beddington, William C. Skarnes, and Marc Tessier-Lavigne. 1996. “Netrin-1 Is Required for Commissural Axon Guidance in the Developing Vertebrate Nervous System.” *Cell* 87(6):1001–14.

Topilko, Piotr, Sylvie Schneider-Maunoury, Giovanni Levi, Amina Ben Younes Baron-Van Evercooren, Anne Chennoufi, Tania Seitanidou, Charles Babinet, and Patrick CHARNAY. 1994. “Krox-20 Controls Myelinisation in the Peripheral Nervous System.” *Nature* 371:796–99.

Yung, A. R., A. M. Nishitani, and L. V. Goodrich. 2015. “Phenotypic Analysis of Mice Completely Lacking Netrin-1.” *Development* 3686–91.

[Editors' note: further revisions were suggested prior to acceptance, as described below.]

While the reviewers agree that some of their concerns have been satisfactorily solved, there are remaining issues that need to be addressed in order for the manuscript to be considered for publication. The reviewers feel that the potential of this study is not fully developed with many opportunities and controls missed. The reviewers emphasize that they do not expect the authors to severely expand the scope of the study, but rather to be as thorough as expected. In light of their comments, addition of the experimental data requested below, will be absolutely essential for further consideration of this manuscript.1. In general, the reviewers find the provided quantifications and its statistical power to be convincing. A remaining concern in this regard relates to the size measurements of the sciatic nerve. For instance, the sciatic nerve in Figure 6 and its S1 is very different between mutant and controls in terms of its branching pattern, thickness etc. This rises some doubts about whether the measurement are done consistently in all animals. The authors need to provide additional information and examples on how the nerve size was measured.

We are grateful to the reviewers to raise this issue. Indeed, in this study we analyzed either dissected sciatic nerves or cleared limbs at different developmental stages. Methodology was adapted depending on stage to be as accurate as possible. From cleared limbs, the same portion of the sciatic nerve, from the node formed by the 3 branches coming from the DRG to the first distal branches was always used for the 3D reconstruction (Figure 6- Figure supp 2 A). In Figure 6 and its S1, we did not analyze the same parameter: at E17,5, when the sciatic nerve is vascularized, we quantified the nerve area and found no significant difference; whereas at E16 (not vascularized) the nerve morphology is different and more branched.

We concluded that measuring the nerve average diameter (reflecting the thickness) was more accurate, and would take into account potential differences between the mutant and control in term of branching and fasciculation. In addition, we added a figure describing how the nerve area was measured (Figure 7) and a corresponding paragraph (**‘**Quantification of the nerve area’) in the MandM section that give precise information for all figures regarding the method used.

2. An additional major problem is that some of the reviewers' concerns have been addressed through the rebuttal letter without any significant or explicit edits in the manuscript. For example, the comment #1 about the specificity of the effects of the Krox20 mutation on myelination gets 5 paragraphs and many references in the rebuttal letter, but only a Results section mention that the sciatic nerves "from Krox20 Cre/Flmice are devoid of myelin" (line 218). Generally, in the rebuttal, the authors use unpublished observations, and references to on-line databases but these are not validated. This should be fully amended in the revised version.

We fully agree and we introduced modifications in the revised version of the manuscript according to the reviewer’s comments. In particular, argumentations concerning the major point 1 in the first rebuttal letter can now be found in the manuscript, in results and discussion (pages 10 and 17). References to online databases have also been now validated by RNAscope and immunostaining data (Figure 6).

3. The identity of netrin1 expressing cells is not convincing and needs further work. The authors added a new result, showing that at E17 (onset of vascularization) the netrin1 signal appears close to the developing vasculature, as opposed to E16.5, where netrin signal is dispersed (supp. Figure 6A, B). They suggest that at P1 Netrin1 is expressed by the epineurial and perineurial cells and proliferating fibroblast like cells, based on a recent scRNA-seq analysis from Gerber et al., 2021. However, the authors didn't provide any validation to this using relevant markers to show the co-expression with these cells. The qPCR quantification of netrin1 receptor expression also falls short of a developmental paper. What is missing is a demonstration of mRNA expression by in situ, for example using RNAscope.

According to the referee’s suggestions we added new data concerning the expression of Netrin-1 and UNC5B at the mRNA and protein levels. Using RNAscope and immunolabelling we confirmed the expression of UNC5B by the endothelial cells and of Netrin-1 by the fibroblasts located at the peri and epineurial layers (Figure 6B, C, C’ ,K, K’ and L).

4. The authors did a good job demonstrating the development of the sciatic nerve vascularization and that the elimination of the Schwann cells/myelination disrupts the regulation of the angiogenesis, however, it is still not clear whether this is a direct effect, or secondary effect due to affected neurons. Netrin1 is required for the normal development of peripheral nerves. Poliak et al. (PMID: 26633881) show this explicitly. Netrin1 mutation (same one as used here) affects the guidance of motor axons within peripheral nerves, including the sciatic nerve. The authors should extend the analysis to additional nerves (e.g., the diaphragm, which is a classic peripheral nerve, or purely sensory/cutaneous nerves) to convincingly argue that the phenotypes are generalizable to different peripheral nerves, as previously suggested by the reviewers.

We thank the reviewer for this comment and understood that there was a concern about (1) sciatic nerve misguidance of motor neurons and (2) sensory nerves vascularization. In the article published by Poliak et al., misguidance of motor axons were observed at E11,5 and E12,5 in the ventral and dorsal (upper/lower) limb region in *Ntn1^Lac/LacZ^* corresponding to the future tibial and peroneal nerves. As explained above, we analyzed (a) at later stages (E16 and E17.5) and (b) the segment of the sciatic nerve anatomically located before the division of the two dorsal and ventral branches of the motor nerves, that was not analyzed in Poliak et al. study. The guidance defects reported by Poliak et al. are aberrant and misguided motor axons along the dorsoventral axis of the limb, but this misguidance affect different sciatic nerve branches than the segment analyzed in our study that is located before those branches. Indeed, we have not identified any significant differences in sciatic nerve thickness at E16 and E17,5 between *Ntn1^+/+^* and *Ntn1^Lac/LacZ^* (Figure 6 D and H, and supplement figure 6 D) in the segment analyzed.

To address the question about the role of Netrin on vascularization of other, mixed or purely sensory nerves we analyzed 3 other nerve types: phrenic nerve, and intercostal (mixed motor/sensory), cutaneous and sural (purely sensory branch of the sciatic nerve). None of those nerves appears vascularized in peri-natal stages (E17,5-P2) (Figure 6- Figure supp 2). Thus, as those nerves are very thin at peri-natal stages and mostly unmyelinated or poorly myelinated in the adult, the timing of vascularization may be delayed making it impossible to study the vascular phenotype in the *Ntn1^Lac/LacZ^* model (because they die before birth). Furthermore, as for the cutaneous and sural nerves, if vascularized, the vascularization of those nerves will be more peri nervous and therefore different from what is observed in the sciatic nerve. In fact, in the adult mouse, the sural nerve, aligned with a vein, is reported to have just a small amount of micro vessels, “often on the nerve’s surface as nerve diameters approach the typical diffusion lengths for molecular oxygen” (Dudele, Rasmussen, and Østergaard 2020).

5. The expression of the suggested Netrin1 receptors (unc5b) by the endothelial cells of the sciatic nerve – the data provided by the authors are not sufficient to suggest that Netrin1 regulates angiogenesis through this receptor. Although this might require the analysis of another transgenic line, or perhaps isolation and culture of ECs from sciatic nerves, this part would be a clear prove of the direct role of Netrin1 in the sciatic vasculature.

To tackle this issue, we chose to perform in vivo inactivation of UNC5B using blocking antibodies to explore its role in the sciatic nerve vasculature. We chose to inactivate UNC5B at P0, a stage when angiogenesis is still ongoing as attested by the presence of angiogenic sprouts (Figure 1M) and UNC5B and Netrin1 are expressed by INV endothelial cells and epi/perineurial cells respectively (Figure 6 A-C’; K-L). We added new data showing that injections of a UNC5B blocking antibody to WT pups at P0 is sufficient to significantly reduce the density of blood vessels inside sciatic nerve (Figure 6M), phenocopying the observations made in *Ntn1^Lac/LacZ^*. Altogether those data support the hypothesis that Netrin1 regulates angiogenesis through UNC5Breceptor*.*

6. Throughout Figure 1 the authors use the "tip cell" term to refer to angiogenic sprouts. However, endothelial tip cells are characterized by the presence of specific markers that the authors do not provide in their characterization. Hence the definition and corresponding quantification should be changed to either "branching points", "end points" or angiogenic sprouts. All these are standardized measurements offered by different image processing tools.

We agree with the reviewers, and we changed the “tip cell” term to “angiogenic sprout”.

Dudele, Anete, Peter Mondrup Rasmussen, and Leif Østergaard. 2020. “Sural Nerve Perfusion in Mice.” *Frontiers in Neuroscience* 14(December):1–12.